# 2D CdPS₃-based versatile superionic conductors

Xin Yu[1,2] & Wencai Ren ●[1,2] ✉

Ion transport in nanochannels is crucial for applications in life science, filtration, and energy storage. However, multivalent ion transport is more difficult than the monovalent analogues due to the steric effect and stronger interactions with channel walls, and the ion mobility decreases significantly as temperature decreases. Although many kinds of solid ionic conductors (SICs) have been developed, they can attain practically useful conductivities (0.01 S cm⁻¹) only for monovalent ions above 0 °C. Here, we report a class of versatile superionic conductors, monolayer CdPS₃ nanosheets-based membranes intercalated with diverse cations with a high density up to ~2 nm⁻². They exhibit unexpectedly similar superhigh ion conductivities for monovalent (K⁺, Na⁺, Li⁺) and multivalent ions (Ca²⁺, Mg²⁺, Al³⁺), ~0.01 to 0.8 S cm⁻¹ in the temperature range of −30 – 90 °C, which are one to two orders of magnitude higher than those of the corresponding best SICs. We reveal that the high conductivity originates from the concerted movement of high-density cations in the well-ordered nanochannels with high mobility and low energy barrier. Our work opens an avenue for designing superionic conductors that can conduct various cations and provides possibilities for discovering unusual nanofluidic phenomena in nanocapillaries.

Developing artificial pore structures with angstrom-scale channels to mimic the ion transport in biological systems is crucial for applications in life science, filtration, and energy storage[1–3]. The rapidly growing diversity and safety concern of electrochemical energy storage technologies calls for solid ionic conductors (SICs) that can conduct various ions in the nanochannels, such as Li⁺, Na⁺, K⁺, Ca²⁺, Mg²⁺, and Al³⁺, which provide ionic current to support the cell reactions[4,5]. In particular, batteries based on multivalent ion transport and storage offer a promising path to reach higher energy density and/or capacity density than current Li-ion batteries, along with the advantages of long-term cost and sustainable resource availability[4–6]. Furthermore, with the continuous upsurge in demand for energy storage, batteries are increasingly required to operate under temperature extremes[7]. Various materials, including inorganic solids, polymers, metal-organic frameworks (MOFs), and two-dimensional (2D) materials have been developed for SICs[2–14]. However, multivalent ion transport in the nanochannels is much more difficult than the monovalent analogs due to the steric effect and stronger interactions with channel walls[2,5,6,8], and ion mobility decreases significantly as temperature decreases[5,7]. Thus, the conductivities of multivalent SICs are at least one order of magnitude smaller than those of monovalent analogs, typically lower than 10⁻³ S cm⁻¹ at room temperature (RT). So far, nearly none of the existing SICs are able to attain practically useful conductivities (0.01 S cm⁻¹) below 0 °C, even for the most common monovalent ions such as Li⁺ and Na⁺.

Recently, we reported a class of membranes assembled from monolayer MPX₃ nanosheets (M = Cd, Mn, Fe, Co, Ni, Zn or Cr; X = S or Se), showing exceptionally high proton and Li⁺ conductivities in the temperature range of 30 to 90 °C[15]. Unlike MOFs and other lamellar membranes, such as graphene oxide (GO) and MXene membranes, the nanochannel walls of MPX₃ membranes contain a huge number of negatively charged transition-metal vacancies[15]. In this paper, we show

[1]Shenyang National Laboratory for Materials Science, Institute of Metal Research, Chinese Academy of Sciences, 72 Wenhua Road, Shenyang 110016, P. R. China. [2]School of Materials Science and Engineering, University of Science and Technology of China, 72 Wenhua Road, Shenyang 110016, P. R. China. ✉e-mail: wcren@imr.ac.cn

that the vacancy-containing $CdPS_3$ nanosheet-based membranes can be intercalated with diverse cations with a high density of up to 2 $nm^{-2}$ for versatile superionic conductors. They exhibit similar superhigh conductivities for monovalent ($Li^+$, $Na^+$, $K^+$) and multivalent ions ($Ca^{2+}$, $Mg^{2+}$, $Al^{3+}$), ~0.01 to 0.8 S $cm^{-1}$ in the temperature range of −30 to 90 °C, which are one to two orders of magnitude higher than those of the corresponding best SICs.

## Results

### Fabrication and characterizations of CdPS₃-Z membranes

We first fabricated Cd-vacancy-containing monolayer $Cd_{0.85}PS_3Li_{0.3}$ nanosheet-based membranes following the approach described previously in ref. 15 (Supplementary Figs. 1, 2). The $Cd_{0.85}PS_3Li_{0.3}$ nanosheet dispersion shows a zeta potential of −51.8 mV (Supplementary Fig. 3), indicating the nanosheets are highly negatively charged. Membranes with other cations were obtained through ion exchange by immersing the $Cd_{0.85}PS_3Li_{0.3}$ membranes into different salt solutions to replace $Li^+$ with other desired cations, including $K^+$, $Na^+$, $Ca^{2+}$, $Mg^{2+}$, and $Al^{3+}$ (Fig. 1a, Methods). The resulting membranes were denoted as $CdPS_3$-Z, where Z represents the cations in the membranes. It should be noted that the ion exchange process of the $CdPS_3$-Li membrane is different from that of the GO-based membrane in which cations can control the interlayer distance by strong cation-π interactions and exclude other cations that have larger hydrated volumes[16]. There seems to be no clear order for the ion exchange process of the $CdPS_3$-Z membrane (Supplementary Figs. 4–6), which we attribute to the weak electrostatic interactions between cations and the surface of the nanochannels[15], similar to the ion exchangeable materials such as vermiculite[17,18].

On the basis of the crystal structure and chemical composition of $CdPS_3$-Li, the surface charge density of the 2D nanochannels was estimated to be on the order of 2.0 $e$ $nm^{-2}$, which is one to two orders of magnitude higher than those reported for carbon and h-BN nanotubes, silica channels, GO laminates, and pores in monolayer crystals, typically ranging from -0.06 to 0.6 $e$ $nm^{-2}$, and three to four orders of magnitude larger than angstrom-scale 2D slits with graphite walls[2]. Such an exceptionally high density of negative charges attracted a huge number of cations from the salt solution to intercalate into the nanochannels while repelled ions of the same charge. Importantly, these membranes have good stability in water, except for the $CdPS_3$-Li membrane (Supplementary Figs. 7, 8). Moreover, the stability of $CdPS_3$-Li membrane in aqueous solutions can be significantly improved by the incorporation of Li-containing salts (Supplementary Fig. 9). Figure 1b and Supplementary Fig. 10 show that all the $CdPS_3$-Z membranes have well-ordered lamellar structures with uniform distribution of corresponding cations, while no Cl was observed within the detection limit of energy dispersive X-ray spectroscopy (EDS). X-ray photoelectron spectroscopy (XPS) measurements suggest the complete exchange of $Li^+$ by other cations with the valence states of Cd, P, and S being maintained in the $CdPS_3$-Z membranes (Supplementary Figs. 4, 11, 12). Inductively coupled plasma atomic emission spectroscopy (ICP-AES) measurements confirm that $Li^+$ has been completely exchanged with the desired cations on the basis of the charge balance principle (Supplementary Fig. 13 and Supplementary Table 1).

The presence of abundant negatively charged vacancies makes $CdPS_3$-Z membranes highly hydrophilic with a contact angle of 31°, 20°, 23°, 32°, 36°, and 65° under ambient conditions for $CdPS_3$-K, $CdPS_3$-Na, $CdPS_3$-Li, $CdPS_3$-Ca, $CdPS_3$-Mg, and $CdPS_3$-Al, respectively (Fig. 1c and Supplementary Fig. 14), enabling easy adsorption of water molecules in the nanochannels (Supplementary Fig. 15). X-ray diffraction (XRD) measurements reveal the nanochannels in $CdPS_3$-Z membranes. Generally, $CdPS_3$ crystals show a sharp XRD peak at 13.5°, corresponding an interlayer distance ($d$) of 0.65 nm (Supplementary Fig. 16). Such a peak was also observed in the membranes under

ambient conditions (Supplementary Fig. 16), which arises from the sheet restacking of some regions in the case of low RH (~30–50%). It is worth noting that at 98% RH, this XRD peak disappears and all the membranes show a sharp XRD peak around 7° and the corresponding second-order diffraction peak around 14° (Fig. 1d). According to these XRD data, $d$ corresponds to 1.30, 1.26, 1.32, 1.27, 1.30, and 1.32 nm for $CdPS_3$-K, $CdPS_3$-Na, $CdPS_3$-Li, $CdPS_3$-Ca, $CdPS_3$-Mg, and $CdPS_3$-Al membranes, respectively (Fig. 1e). Taking into account the $d$ in $CdPS_3$ crystal of 6.5 Å, these values yield 2D nanochannels with a height ($h$) of ~6.1 to 6.7 Å (Fig. 1e). Note that $h$ is comparable to or much smaller than the hydrated diameter of intercalated cations ($D_H$) (Fig. 1e), thus these ions shredded or flattened their hydration shells when entering and staying in the confined 2D nanochannels[2,19,20].

Note that in comparison with the other membranes, the adsorbed water in the $CdPS_3$-Al membrane is easier to be removed at low RH, possibly due to the much stronger electrostatic attraction between $Al^{3+}$ and vacancies (Supplementary Fig. 17a). The reduced $d$ further increases the electrostatic attraction between $Al^{3+}$ and vacancies, which leads to the re-occupation of vacancies by $Al^{3+}$ along with the removal of hydration shells and consequently the decrease in the number of surface charges. Thus, the $CdPS_3$-Al membrane shows a larger contact angle than other membranes under ambient conditions, as shown above, but it shows a similar contact angle to other membranes at high RH (Supplementary Fig. 17b).

### Ion transport in CdPS₃-Z membranes

To investigate the ion transport within the 2D nanochannels of $CdPS_3$-Z membranes, two-probe alternating current impedance was measured at different temperatures in the range of −40 to 90 °C and 98% RH. Notably, the impedance plots display a semicircle and a spike in the high-frequency and low-frequency regions, respectively (Supplementary Figs. 18–23), which are characteristics of pure ionic conductors[9]. The ionic transference numbers ($t_{ion}$) of $CdPS_3$-Z membranes were determined to be above 0.99 at 30 °C (Supplementary Fig. 24). The ion conductivities $σ$ were calculated by the following equation

$$σ = L/RS \quad (1)$$

where $L$, $S$, and $R$ are the length, cross-sectional area, and resistance of the membranes, respectively. It is important to note all the membranes show similarly superhigh conductivities for monovalent ($Li^+$, $Na^+$, $K^+$) and multivalent ions ($Ca^{2+}$, $Mg^{2+}$, $Al^{3+}$) at the same temperature (Fig. 2). The ion conductivities are in the range of 0.43–0.78 and 0.17–0.32 S $cm^{-1}$ at 90 and 30 °C, respectively, which are dozens of times higher than the practically useful values (0.01 S $cm^{-1}$) (Fig. 2b). The monovalent $K^+$ with the smallest $D_H$ has the highest ion conductivity and the ion conductivities only show a slight decrease as $D_H$ and charges of cations increase (Fig. 2b). Even for the multivalent ions with $D_H$ much larger than $h$, including $Ca^{2+}$, $Mg^{2+}$ and $Al^{3+}$, the conductivities still can reach very high values over 0.43 and 0.17 S $cm^{-1}$ at 90 and 30 °C, respectively (Fig. 2b). This is in sharp contrast to the reported ionic conductors[2,5,6], where the conductivities of multivalent ions are suppressed by over ten times than those of monovalent ions. Another remarkable characteristic is that our membranes still maintain practically useful conductivities even at temperatures as low as −30 °C for $K^+$, $Li^+$, and $Ca^{2+}$ and −20 °C for $Na^+$, $Mg^{2+}$, and $Al^{3+}$ (Fig. 2b).

Figure 3 and Supplementary Tables 2–7 further illustrate the advantages of $CdPS_3$-Z membranes over the existing ionic conductors, including liquid, quasi-solid and solid electrolytes. Generally, liquid and hydrogel ionic conductors can reach high ion conductivity over 0.1 S $cm^{-1}$ at RT. However, the conductivity decreases sharply when the temperature is lower than the freezing point of water. For example, aqueous $Na_2SO_4$ (1 M) shows ion conductivity around 0.12 S $cm^{-1}$ at RT but less than -$10^{-4}$ S $cm^{-1}$ at a temperature below 0 °C. Although SICs

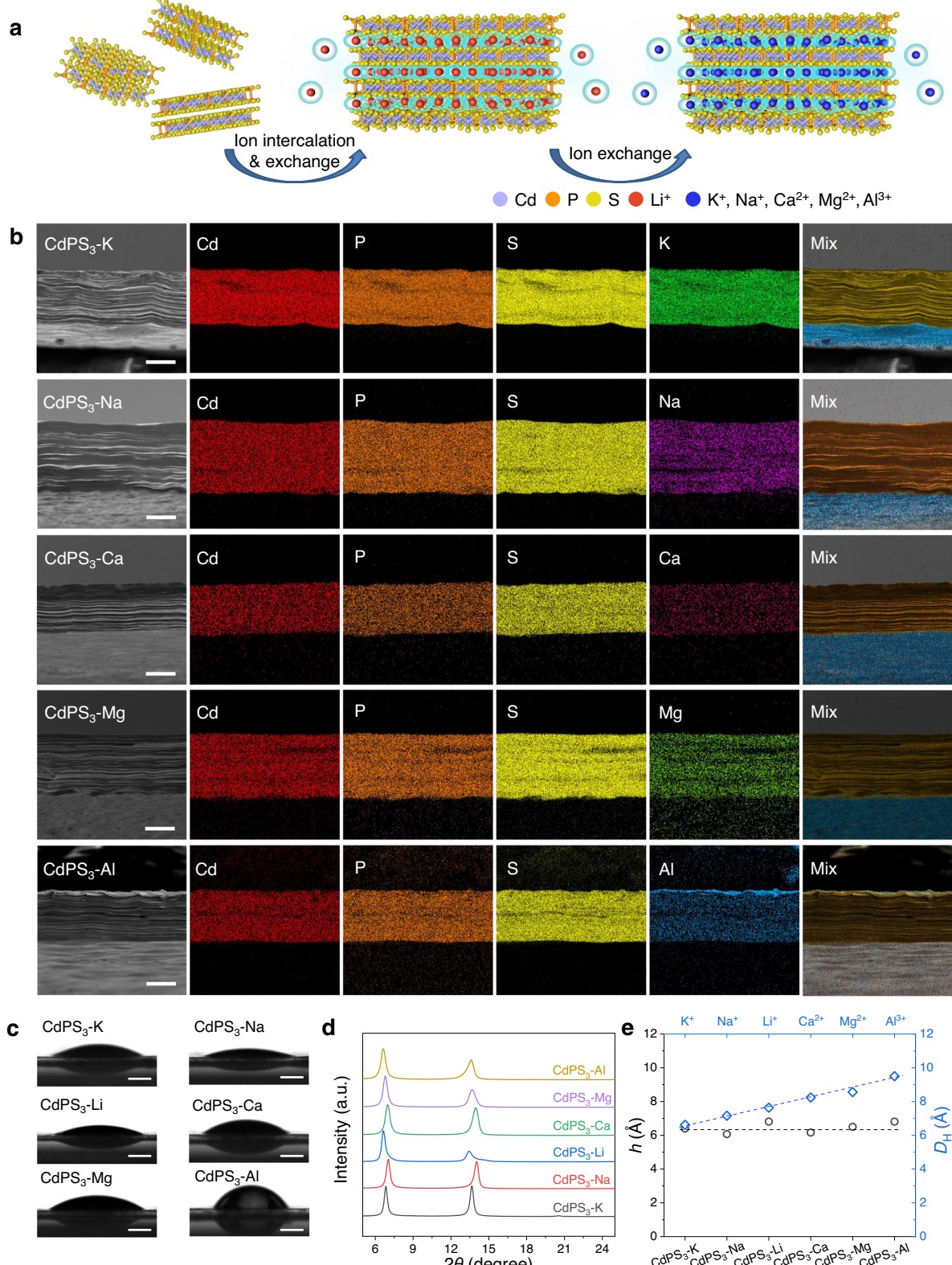

**Fig. 1 | Synthesis and characterizations of CdPS3-Z membranes. a** Schematic illustration of the synthesis process of CdPS$_3$-Z membranes. **b** Scanning electron microscopy (SEM) images and the corresponding EDS elemental mappings of CdPS$_3$-Z membranes. Scale bars, 10 μm. **c** Wettability of CdPS$_3$-Z membranes toward the water. Scale bars, 500 μm. **d** XRD patterns of CdPS$_3$-Z membranes measured at 98% RH. **e** Comparison of $h$ of the 2D nanochannels in CdPS$_3$-Z membranes with $D_H$. Dashed lines are guides for the eye.

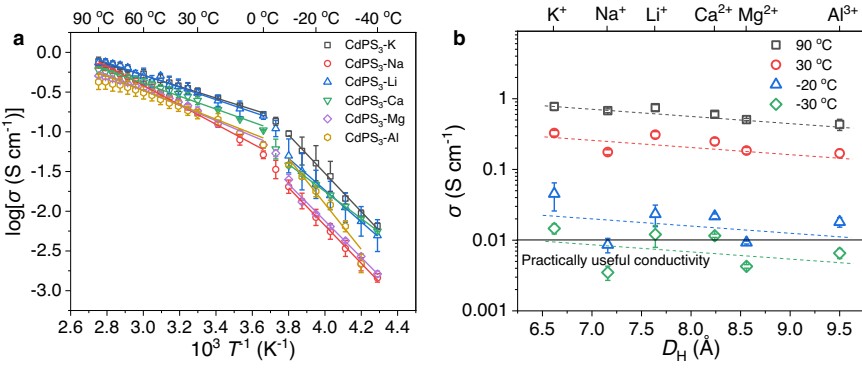

**Fig. 2 | Ion transport properties of CdPS3-Z membranes. a** Arrhenius conductivity plots of CdPS₃-Z membranes measured in the temperature range of −40 to 90 °C and 98% RH. **b** Conductivities of CdPS₃-Z membranes measured at 90, 30, −20, and −30 °C and 98% RH. Dashed lines are guides for the eye, and the practically useful conductivity (0.01 S cm⁻¹) was given for comparison.

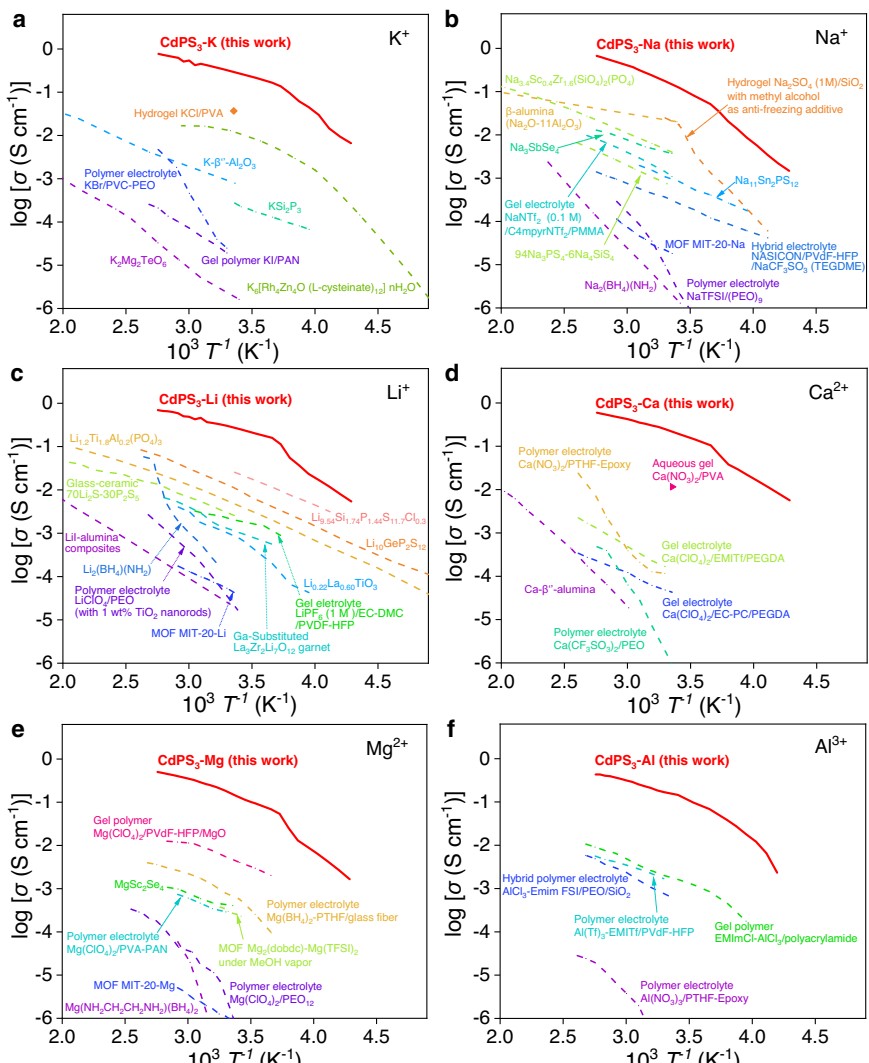

**Fig. 3 | Comparison of ion conductivities of CdPS3-Z membranes with the existing ionic conductors. a–f** Thermal evolution of ion conductivities of CdPS₃-K (**a**), CdPS₃-Na (**b**), CdPS₃-Li (**c**), CdPS₃-Ca (**d**), CdPS₃-Mg (**e**), and CdPS₃-Al membranes (**f**), together with the reported ionic conductors (details see Supplementary Tables 2–7).

do not show a serious conductivity decay as the temperature decreases, their conductivities are lower than the practically useful values below 0 °C as well because of the low baseline conductivities. For multivalent cations, including Ca²⁺, Mg²⁺, and Al³⁺, no SICs reaching the practically useful conductivity are available at RT. Even for the well-known divalent β″-aluminas SICs, the conductivities are on the order of 10⁻⁶ S cm⁻¹ at 40 °C (Ca²⁺, Zn²⁺, Sr²⁺, Ba²⁺, and Pb²⁺)[6]. To the best of our knowledge, the ion conductivities of our CdPS₃-Z membranes are the

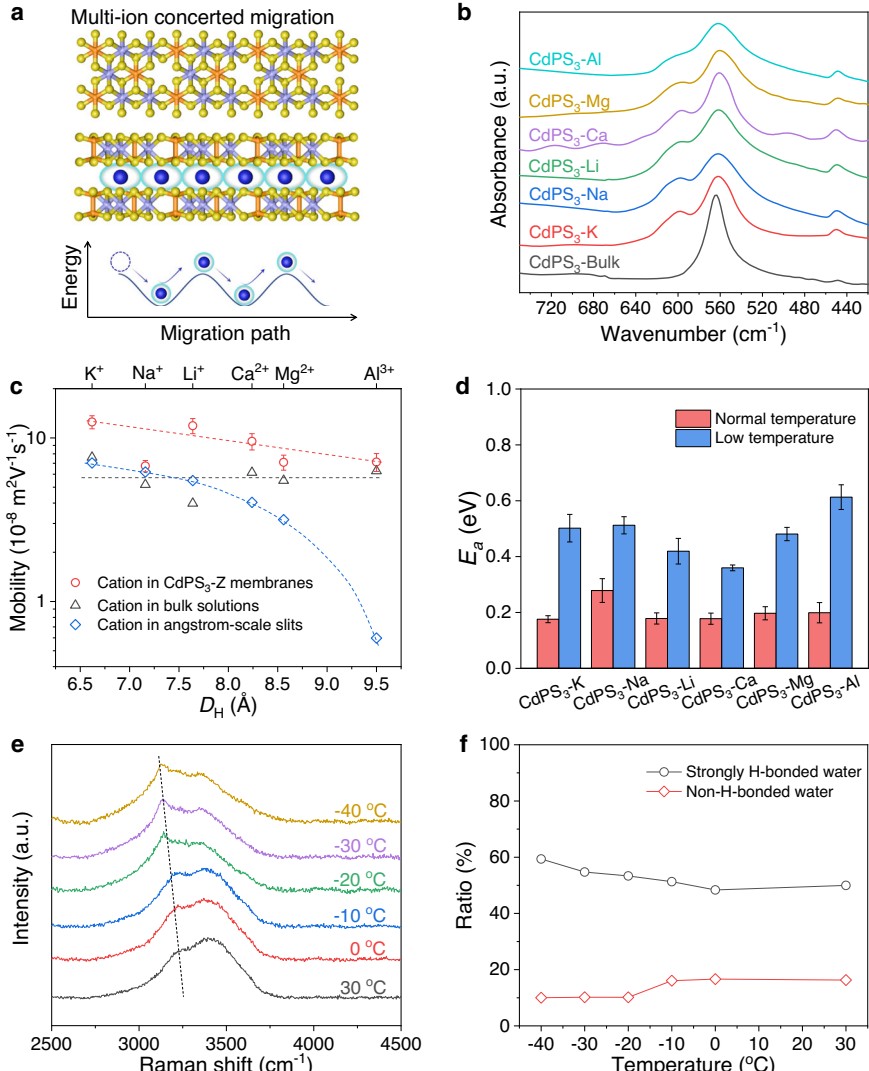

**Fig. 4 | Ion transport mechanism of CdPS3-Z membranes. a** Schematic illustration (top) and the corresponding energy landscape (bottom) for the concerted migration of multiple hydrated ions along the direction of applied voltage in CdPS$_3$-Z membranes. Here, monovalent cations were used as an example, and the in-plane atomic structure (middle) of the CdPS$_3$ layer shows the positions of Cd vacancies. **b** FTIR spectra, showing peak splitting in CdPS$_3$-Z membranes. **c** Ion mobilities in CdPS$_3$-Z nanochannels as a function of $D_H$ of cations at 30 °C. Values for the bulk solutions and the angstrom-scale 2D slits (~6.6 Å in $h$) at RT were presented for comparison[2]. Dashed lines and curves are guides for the eye. **d** $E_a$ at normal- and low-temperature stages for CdPS$_3$-Z membranes. Error bars represent s.d. **e** Raman O-H stretching bands at different temperatures for CdPS$_3$-Li membranes. **f** The corresponding proportion of strongly H-bonded water and non-H-bonded water.

highest among all kinds of ionic conductors, which are several times higher than those of aqueous salt-based ionic conductors including aqueous solutions, water-in-salt electrolytes and hydrogels[21,22], and at least one order of magnitude higher than those reported for practical organic liquid electrolytes, quasi-/solid-state polymers, and inorganic solid electrolytes over a broad range of temperatures from −40 to 90 °C[5,23]. We also investigated the ion conductivities of CdPS$_3$-Z membranes under low RHs (from 30% to 75% RH) at 60 °C. As shown in Supplementary Fig. 25, at comparable ambient humidity (~55% RH), the conductivity of the membranes ranges from about 0.24 to 0.02 S cm$^{-1}$, which is still higher than the practically useful conductivity.

## Ion transport mechanism of CdPS$_3$-Z membranes

To explain the remarkable ion transport behavior in CdPS$_3$-Z nanochannels, we use the concerted ion migration model previously suggested to account for the fast Li$^+$ diffusion in superionic conductors[24], in which concerted migration is activated at high mobile-ion concentration and in unique mobile-ion configuration with high-energy site occupancy enabled by Coulomb repulsion between adjacent ions

(Fig. 4a). Fourier transform infrared (FTIR) spectra show that the asymmetric stretching band of PS$_3$ is split into two absorption bands for all the CdPS$_3$-Z membranes (Fig. 4b), which verifies the presence of Cd vacancies in CdPS$_3$ layers[15]. Thus, individual layers in CdPS$_3$-Z membranes have two types of regions: charge-free pristine CdPS$_3$ and negatively charged Cd vacancies, which construct high-energy and low-energy sites for cation occupancy, respectively, in the 2D nanochannels (Fig. 4a). At a low charge density, the cations tend to gather around the low-energy sites by electrostatic attraction, thus a high energy barrier need to be overcome for cations' transport, and this barrier is significantly increased for multivalent cations with more charges. We estimated the ion density on the basis of the chemical composition of CdPS$_3$-Z membranes, which yielded ~2, 1, and 0.7 nm$^{-2}$ for monovalent (K$^+$, Na$^+$, Li$^+$), divalent (Ca$^{2+}$, Mg$^{2+}$), and trivalent (Al$^{3+}$) ions, respectively. Notably, such ion densities are quite high and compare well with the measurements in the electrochemical charging of bilayer graphene[25], reduced GO flake[19], and ultrathin graphite[26]. The high packing density of cations within the nanochannels could impart strong repulsive Coulomb interaction among adjacent cations to

overcome the electrostatic attractions from vacancies and enable stable high-energy site occupancy, which promotes concerted movements of multiple ions. The ions located at the high-energy sites migrate downhill, which cancels out a part of the energy barrier felt by other uphill-climbing ions (Fig. 4a), leading to a low migration energy barrier and high mobility. The presence of hydration shells also plays an important role in this process since it prevents the cations from being trapped in the vacancies, the low-energy sites.

To support our interpretation, the mobilities of cations in the 2D nanochannels of CdPS$_3$-Z membranes at different temperatures were estimated based on the equation

$$\sigma \approx F(c_+\mu^+ + c_-\mu^-) \qquad (2)$$

where $F$ is Faraday constant, and $c_+/c_-$ and $\mu^+/\mu^-$ are the concentration and mobility of cations/anions, respectively. As mentioned above, no anions' entry was allowed, and thus $c_-$ equals 0. This is different from bulk solutions and chemically inert carbon nanotubes/2D nanoslits, where cations and anions would simultaneously move along opposite directions to generate ionic current[2,27]. Notably, all the cations confined in 2D nanochannels show higher mobilities than bulk solutions with a weak dependence on $D_H$ and charges of cations (Fig. 4c and Supplementary Fig. 26). At 30 °C, the mobility of K$^+$ is ~1.64 times higher than the bulk value at RT. This value is lower than the prediction by mean-field theory (15-fold improvement), but it is reasonable given that the mean-field model assumes cations inside a smooth, defect-free nanochannel[28], and does not take into account the abundant negatively charged vacancies on the channel walls of CdPS$_3$-Z membranes, which are a hindrance to ultrafast ion transport. In particular, the confined Al$^{3+}$ shows mobility ~1.03 times higher than that of the bulk solution, although it has three times more charges with $D_H$ 1.4 times larger than $d$. In sharp contrast, the mobility of Al$^{3+}$ in angstrom-scale charge-free slits of similar $h$ is substantially decreased by ~10 times compared to its bulk mobility due to the steric effect[2]. Furthermore, the mobilities of cations in our membranes are over two orders of magnitude higher than those of the well-known Li$^+$ superionic conductor Li$_{10}$GeP$_2$S$_{12}$ (refs. 9,24). Even at −30 °C, they still maintain values about one order of magnitude higher than the latter at RT (Supplementary Fig. 26). Thus, our membrane shows one to two orders of magnitude higher conductivity in the temperature range of −40 to 90 °C despite of much lower Li-ion concentration. The screened electrostatic attraction by the hydration shells of cations inside the well-ordered nanochannels of CdPS$_3$-Z membranes may play an important role in their extraordinarily higher mobility compared to Li$_{10}$GeP$_2$S$_{12}$ (ref. 6).

Molecular dynamics simulations previously suggested that as $D_H$ is larger than $h$, the ions under quantum confinement have to overcome the hydration energy to reconfigure their hydration shells that become effectively squashed[2]. The hydration energy is on the order of 3–50 eV for the ions with $D_H$ between K$^+$ and Al$^{3+}$ (ref. 29). It has been experimentally observed that $\mu^+$ values evolve approximately exponentially with the hydration energy[2,20], which is in conceptual agreement with the dependence of ions' conductivities on the activation energy barrier $E_a$. We calculated $E_a$ for ion transport in CdPS$_3$-Z membranes by using the equation

$$\ln(\sigma T) = \ln\sigma_0 - \frac{E_a}{RT} \qquad (3)$$

where $\sigma_0$ is a preexponential factor, $R$ is the gas constant, and $T$ is temperature. The evolution of ion conductivity with temperature is divided into two stages: normal-temperature stage (90–0 °C) and low-temperature stage (−10 to −40 °C) (Supplementary Fig. 27). As shown in Fig. 4d, the fitted $E_a$ for K$^+$, Na$^+$, Li$^+$, Ca$^{2+}$, Mg$^{2+}$, and Al$^{3+}$ transport at the normal-temperature stage is 0.18 ± 0.01, 0.28 ± 0.04, 0.18 ± 0.02,

0.18 ± 0.02, 0.20 ± 0.02, and 0.20 ± 0.04 eV, respectively. These values are very similar and compare well with the migration barrier for concerted migration of K ions in biological channels (<0.2 eV)[30] and Li ions in superionic conductors (~0.2–0.4 eV)[24], further demonstrating the significant role of Coulomb repulsion between high-density cations in accelerating ion transport inside the nanochannels. At a low-temperature stage, a relatively larger $E_a$ reveals the increase in the ions' migration barrier. Fortunately, it is still small enough (0.36–0.61 eV), about one order of magnitude lower than that of frozen liquid electrolyte[31].

Taking CdPS$_3$-Li membranes as an example, Raman spectra were recorded to reveal the structure evolution of water molecules in the 2D nanochannels in the temperature range of −40 to 30 °C at 98% RH. Generally, the O-H stretching vibration of water molecules shows a broad peak at 3000–3700 cm$^{-1}$, which is composed of three components related to different intermolecular bonding degrees of water[31,32]. For pure water, the proportion of strongly H-bonded water (at ~3230 cm$^{-1}$) greatly increases while non-H-bonded water (at ~3600 cm$^{-1}$) disappears as the temperature decreases below 0 °C (Supplementary Figs. 28, 29a, b), suggesting the formation of ordered ice. However, the proportion of strongly H-bonded and non-H-bonded water molecules in CdPS$_3$-Li membranes only slightly increases and decreases, respectively, even down to −40 °C (Fig. 4e, f and Supplementary Fig. 29c, d). The H-bond is formed between the partially positively charged H atoms and partially negatively charged O atoms of neighboring H$_2$O mainly by electrostatic interaction[31]. Taking into account the high-density cations confined within the 2D nanochannels, which contain at most bilayer water, we expect that the coordination of water molecules with cations would destroy the H-bond interactions between water molecules. The destruction of H-bonds enables the survival of non-H-bonded water molecules, which ensures practically useful conductivities even down to −30 °C. This is in stark contrast to the increased freezing point for confined pure water molecules in 2D nanochannels, where the van der Waals pressure leads to water-to-ice transformation at RT[33].

## Discussion

Our work demonstrates that the high-density cations inside the well-ordered 2D nanochannels of CdPS$_3$-Z membranes enable concerted ion migrations, which overcome the steric effect, strong interaction with channel walls, and freezing of aqueous solutions that limit the migration of cations in nanochannels. Thus, the membranes exhibit superhigh conductivities for both monovalent and multivalent ions over a broad range of temperatures (−30 to 90 °C), which are over one order of magnitude higher than those of the corresponding best SICs. These findings not only provide insights into the nanofluidic phenomena in the confined nanocapillaries but also open an avenue for designing superionic conductors that can conduct various cations under temperature extremes, which are essential for the development of diverse electrochemical devices with expanded applications.

## Methods

### Synthesis of CdPS$_3$-Z membranes

Cd$_{0.85}$PS$_3$Li$_{0.3}$ nanosheets with Cd vacancies were synthesized by a two-step ion intercalation and exchange method using CdPS$_3$ crystals as raw material[15]. In brief, 0.1 g of CdPS$_3$ crystals were stirred in a 10 mL solution of 1 M KCl, 2 M K$_2$CO$_3$, and 1 M EDTA at 55 °C for 3 h and then filtered and washed with deionized water several times. The as-obtained product was mixed with 1 M LiCl at 35 °C for 2 h, followed by repeated washing to remove residual salts. The resulting product was redispersed in deionized water and sonicated for several minutes to facilitate exfoliation. After centrifuging at 10,000 rpm for 5 min to remove unexfoliated crystals, the stable Cd$_{0.85}$PS$_3$Li$_{0.3}$ dispersion obtained was filtered through polycarbonate membranes in a vacuum filtration cell to fabricate Cd$_{0.85}$PS$_3$Li$_{0.3}$ (CdPS$_3$-Li) membranes.

To obtain other CdPS$_3$-Z membranes, CdPS$_3$-Li membranes were immersed in 0.5 M aqueous chloride solution of desired cations (K$^+$, Na$^+$, Ca$^{2+}$, Mg$^{2+}$, Al$^{3+}$) for more than 48 h, and then thoroughly rinsed with deionized water before measurements and characterizations. During this process, Li ions were completely exchanged with desired cations (Supplementary Table 1).

## Structure characterizations

The thickness and lateral size of the exfoliated nanosheets were measured using an atomic force microscope (AFM, Bruker Multimode 8). The Zeta potential of Cd$_{0.85}$PS$_3$Li$_{0.3}$ nanosheet suspensions were measured on Malvern Zetasizer Nano-ZS90. The morphology and thickness of CdPS$_3$-Z membranes were characterized by SEM (Verios G4 UC, 10 kV). The cross-sectional element distribution of the membranes was analyzed by EDS (Oxford Instrument Ultim Max 100). XPS measurements were conducted on ESCALAB 250 with an excitation source of Al Kα X-ray (1486.6 eV). The elemental compositions of the membranes were characterized by ICP-AES (PerkinElmer, OPTIMA 8300DV). More than three samples were measured to determine the chemical composition of each material. The hydrophilicity of the membranes was evaluated by a contact angle analyzer at 298 K (Dataphysics, OCA20). FTIR spectra were measured using a UV-vis absorption spectrometer (JACSO V-550).

The nanochannels of CdPS$_3$-Z membranes at 98% RH were characterized by XRD (Rigaku, D/MAX 2400 and PANalytical, X'Pert PRO using Cu Kα radiation). CdPS$_3$-Z membranes were first taken out from the salt solution and washed with deionized water. After the surface water was removed, the CdPS3-Z membranes were stored in a sealed homemade container with 98% RH for overnight to make sure that the membranes were under equilibrium conditions. The measurements were carried out at RT as quickly as possible to avoid the dehydration of the membranes.

To reveal the structural evolution of water molecules in the nanochannels of CdPS$_3$-Li membranes with temperature, Raman measurements were performed in the temperature range of −40 to 30 °C using a Raman spectrometer (Witec alpha 300 R, excited by 532 nm laser). Before measurements, the membranes were kept at 98% RH for over 48 h to ensure complete hydration. During the measurements, the CdPS$_3$-Li membrane was sandwiched between two quartz plates to minimize the change in the humidity of the membrane. The ratio of strongly H-bonded and non-H-bonded water was calculated based on the areas of fitted peaks.

## Measurements of ion transport

All the samples used in our study had a dimension of 15 mm × 5 mm (length × width) and slightly varied thickness, which was determined by using cross-sectional SEM imaging. Before measurements, the membranes were fixed on a homemade device with two sides attached to platinum electrodes. To ensure that the samples reached a near-equilibrium state, they were kept at room temperature in a 98% RH environment overnight for 98% RH and variable temperature experiments. For 60 °C and variable humidity experiments, we found no significant changes in conductance after exposing samples to a certain RH for 2 and 20 h at 60 °C. Therefore, samples were exposed to various humidity environments for over 2 h before testing. The impedance measurements were carried out on Autolab electrochemical workstation (PGSTAT204) with frequencies ranging from 20 Hz to 1 MHz and an alternating potential of 150 mV. The conductivities at temperatures from 30 to 90 °C and 98% RH were tested in a constant temperature and humidity chamber (Shenzhen Hongjian Instruments Co. Ltd), and those from −40 to 30 °C and 98% RH, as well as low RH experiments were tested on a constant temperature and humidity chamber (Votschtechnik, LabEvent). At least three samples were measured for evaluating the ion conductivities for each kind of membrane.

## Estimation of ionic transference number

Chronoamperometry was used to differentiate the electron and ion conductance in the membranes at 98% RH and 30 °C. This electrochemical technique measures the current as a function of time by applying a constant potential to the electrodes. An inert Pt electrode was used to block ion transport in our experiments. As shown in Supplementary Fig. 24a, after the device was subjected to a direct voltage, ions accumulated at the interface between the tested membrane and Pt electrode, creating a significant internal inverse electric field that inhibited ion transport. As ions were blocked, the current was primarily carried by electrons, allowing us to distinguish between the ion and electron contributions. At the initial stage of measurement, the current ($i_T$) was derived from both ionic and electronic currents. However, in the static state, the ionic conductance was blocked and only the electronic current was measured ($i_e$) (Supplementary Fig. 24b). Thus, the $t_{ion}$ was calculated as follows:

$$t_{ion} = \frac{i_T - i_e}{i_T}$$

Note that the $t_{ion}$ were nearly identical regardless of the polarity of the applied direct voltage.

## Data availability

The authors declare that the experimental data supporting the results of this study can be found in the paper and its Supplementary Information file. The detailed data for the study is available from the corresponding author upon request.

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

## Acknowledgements

This work was financially supported by the National Natural Science Foundation of China (Nos. 52188101), the Key Research Program of Frontier Sciences of the Chinese Academy of Sciences (No. ZDBS-LY-JSC027), and the Strategic Priority Research Program of the Chinese Academy of Sciences (No. XDB30000000). The authors thank X. T. Qian and Z. B. Liu for their help on experiments and valuable discussions.

## Author contributions

W.R. conceived and supervised the project. X.Y. performed the experiments. W.R. and X.Y. designed the experiments, analyzed the data, and wrote the manuscript.

## Competing interests

The authors declare no competing interests.
