## [Peer Review File · Nature Communications]

2D CdPS3-based versatile superionic conductorsREVIEWER COMMENTS

Reviewer #1 (Remarks to the Author):

The manuscript by Yu and Ren reports a 2D CdPS3-based membrane that exhibits high ion conductivity for monovalent and multivalent ions from -30 to 90 °C. The rapid transport at low temperatures is attributed to the concerted movement of cations and the presence of the non-H-bonded water molecules. The reported results are very interesting and novel. Therefore I recommend the publication of the manuscript after the authors have taken care of the following minor comments.

1. For the experiment, the authors applied 95% RH for all measurements to ensure the cations were always hydrated. Does the author check the ion transport at lower or even 0% RH?
2. The polarization plot measured by the authors was swept towards only one direction (Sup Fig.15). What is the performance of the membrane under the cyclic voltammetry? Is the rapid transport of cations reversible?
3. The fast permeation of cations is a very interesting phenomenon. But the authors need to comment on the stability of the cation-exchanged membrane. Do the cations get replaced while using the membranes in different conditions?

Reviewer #2 (Remarks to the Author):

In the paper the authors have described a super-ionic conductor i.e. CdPS3 based membranes for the applications in wide temperature range from -40 to 90 degree Celsius. The authors have observed that the ionic conductivity of these membranes is highest till reported. The main point of the paper is the highest membrane conductivity at temperatures below zero degree Celsius. The authors observe that for their membranes, the ionic conductivity remains more or less the same (line 115) up to -40 degree Celsius. While the observations are very interesting from a fundamental point of view, there are many grey areas, which needs to be addressed before making any decision on the publication.

Using various characterizations, the authors have found that upon intercalation, the interlayer spaces of these membranes are occupied by cations in large numbers exceeding previous reports (line 154) and the ions exhibit mobilities much larger than the bulk (line 174). The authors have given a model based on concerted ion migration for explaining the high values of conductivities (line 144-163). It basically states that the negatively charged Cd vacancies attract lots of cations to the interlayer spaces and the large number of cations move together and have a low ion migration energy barrier that give rise to high mobility.

This point needs careful consideration. The cartoon provided in figure 4a suggests one-dimensional transport, however, two-dimensional ion-migration picture should lead to a lower mobility. It is not clear as how the hopping rates are enhanced in this picture. Further, Al³⁺ mobility is similar to bulk, and there is no reason to think that there is any hydration picture that needs to be invoked either. The interlayer space do not show any trend with cations, indicating that hydration radius is not a good parameter to discuss. The authors should have tried aqueous salt solution-based conductivity experiments to understand the nature of the walls as all of the experiments were done at 98% humidity. In addition, the authors also discuss the stronger interaction of water with cations to explain the decreased freezing point of water. The stronger interaction would lead to a hydration dominated transport, which the authors are unable to observe. The contact angle is pointing to a different picture for Al³⁺, with a contact angle of 63 degrees. The less hydrophilicity would suggest a reduced surface charge, leading to less electrostatic interactions with the ions and hence recovering to the bulk mobility for Al³⁺. This also suggests that electrostatic interactions can fully explain the experimental result.

According to the Raman characterization by the authors, below the freezing temperature, the structure of water inside the interlayer spaces is not similar to bulk ice (supplementary fig 18). It has a lower number of strong hydrogen bonds and is not in the completely ordered state as that of bulk ice. Also, there are just two layers of water molecules inside the interlayer space and most of them are interacting with cations, leaving no place for hydrogen bonds between water molecules. Since there is no change in the state of water, mobility remains fairly constant.

1. The percentage of cations is calculated from ICP-AES, and the absence of chlorine is confirmed through EDAX. Although XPS has been done, it is not used to calculate the percentage of ions or absence of chlorine in the membrane. A survey spectra of XPS with depth analysis is also missing which would have been a better proof for the distribution and estimating the percentage of ions intercalated.
2. The large negative charge of the layers would make the layers to be very unstable, especially in water related applications. The large negative charge of the layers is a concern for fouling also. Did the authors tried any experiment to test the stability of these membranes in water?
3. Estimation of mobility can be erroneous as it has been estimated from the conductivity.
4. The estimation of conductivity is geometry dependent and the authors need to provide the actual parameters that are used for the estimation.
5. It is not clear how the authors estimated transference number from the chronoamperometry data. It needs to be explained atleast in the supplementary text.
6. The activation energy plot clearly shows two slopes. At low temperatures, the energy barrier for ion migration is slightly higher. The activation energy mentioned for the intercalants do not show any trend or the variations are within the error bar. However, the mobility clearly shows a decreasing trend with increase in valence, which is similar to the trend in contact angle, atleast for the end points.

7. There is no mention of how many samples showed similar behavior and what is the percentage of samples that showed the same trend. Error bars are missing from the plots.
8. The contact angle of CdPS3-Li is lower than CdPS3-Na. Ideally, Li should be more hydrophilic than Na. Is the observed contact angle within the error bar?
9. The studies are done at 98% RH. The stability of the membranes if dipped in water, is not discussed at all.
10. In line 80, it is mentioned that abundant cations make the membrane hydrophilic, so why is aluminum less hydrophilic than the rest? How is the charge playing a role here in hydrophilicity? Contact angle clearly shows a large variation.
11. For XRD, in line 88, it is mentioned that there is sheet restacking at low relative humidity, due to which there is a bulk peak (13.2 degrees) at low RH; can this be plausibly due to incomplete exchange? Apart from this, the extra peaks above 15 degrees at low RH need to be properly accounted for.
12. On line 89, XRD peak at 13.5 degree mentioned to disappear, however, in the figure it is still visible.
13. y-axis scale of Supplementary Fig 15 a) can be represented better, e.g. from 0-20 so that the variation can be more evident.
14. Line 119-Is -30 deg a typo? Is it meant to be 30 deg? Otherwise -30 deg is not in the Fig. Lowest conductivity temperatures are different for different cations. Is there any reason?
15. Some references can be given for lines 135-138.

Reviewer #3 (Remarks to the Author):

This work investigates the conductivity of 2d materials with high density of ion intercalation with the aim to develop highly conductive solid ionic conductors. They report a class of versatile superionic conductors, monolayer CdPS₃ nanosheets-based membranes intercalated with diverse cations with a high density. The high conductivity is even demonstrated under a temperature as low as -30 degree C. Their finding benefits the design of solid ionic conductors. However, I have following questions and comments.

- (1) The authors use an immersion method to replace Li⁺ with other ions inside the 2d layer. It is known that Li⁺ has lower hydration radius and thus lower dehydration energy than multivalent ions. It is counter-intuitive that those multivalent ions can replace Li ions inside the 2D channels. As shown in the paper of [Nature 550, 380-383 (2017)], the weak cation- π interaction of potassium ions can prevent the intercalation of ions with large hydration radius. This work claims the surface charge density as high as 2 e/nm². The electrostatic interaction between 2D layers and cations should be much higher and more stable, rendering it difficult for lithium ions to be replaced by multivalent ions. Thus, it is necessary to analyze the energy barrier for the ion replacement.
- (2) The authors claim the surface charge density of Cd_{0.85}PS₃Li_{0.3} is on the order of 2 e/nm² based on the composition analysis. Does surface zeta potential measurement support this argument?
- (3) In fig.2, the energy barrier for ions movement inside the 2d channels can be extracted from the Arrhenius plots. Why does the sodium ion show the highest energy barrier in the high temperature range, and the Potassium ion show the second largest energy barrier in the low temperature range, as told from the slopes?
- (4) The ions inside the 2D layers are still hydrated, or at least partially. Can the authors explain the mechanism of high conductivity of the ionic conductor under a temperature of -30 degree C?
- (5) The authors use concerted ion movement to explain the high mobility of ions inside the 2d channels. Can the enhancement of 1.6 times higher mobility of K⁺ be explained by the mean field theory with considering the Coulomb interaction of positive ions? In ref. 16, the mean field theoretical model predicts an enhancement around 40 times in graphene channels.
- (6) In the XRD measurement of Al³⁺ intercalated samples, the peak is very weak. Does it mean the Al³⁺ is hardly intercalated inside the 2D channels? Do the lithium ions completely replaced by other ions during the immersion? How to prove that?

Response to reviewers' comments

Reviewer #1 (Remarks to the Author):

The manuscript by Yu and Ren reports a 2D CdPS₃-based membrane that exhibits high ion conductivity for monovalent and multivalent ions from -30 to 90 °C. The rapid transport at low temperatures is attributed to the concerted movement of cations and the presence of the non-H-bonded water molecules. The reported results are very interesting and novel. Therefore I recommend the publication of the manuscript after the authors have taken care of the following minor comments.

Reply: We thank the reviewer very much for the positive comments and all the valuable and constructive suggestions that have helped us greatly improve the quality of our manuscript.

1. For the experiment, the authors applied 95% RH for all measurements to ensure the cations were always hydrated. Does the author check the ion transport at lower or even 0% RH?

Reply: We thank the reviewer very much for the valuable suggestion.

We have followed the reviewer's suggestion to evaluate the ion transport properties of CdPS₃-Z membranes at 75%, 55%, and 30% RH at 60 °C. As shown in Fig. R1, the conductivity of the membranes decreases with decreasing RH but still much higher than the practically useful conductivities (0.01 S cm⁻¹) even at 55% RH. Moreover, similar to the results at 98% RH, all the membranes show similar conductivity at the same low RH except for CdPS₃-Al membrane. In our previous work, we found that less water molecules are inserted into the nanochannels of CdPS₃-Li membranes at low RHs, leading to a decrease in interlayer distance below the hydrated diameter of intercalated cations [*Science* 370, 596-600, 2020]. Thus, more energies are required for the ions under quantum confinement to overcome the hydration energy to reconfigure their hydration shells that become more squashed. Meanwhile, the electrostatic attraction between the vacancies and cations will increase. As a result, the conductivity of the membranes decreases with decreasing RH. For the CdPS₃-Al membrane, Al³⁺ ions will be attracted into the vacancies on the CdPS₃ nanosheets at low RH (30-50% RH) (Fig. R2). The significantly increased electrostatic attraction suppresses the ion transport, leading to a dramatic decrease in ion conductivity.

We have added the ion conductivities at low RHs and related discussions in the revised manuscript.

Fig. R1 RH-dependent conductivity of CdPS₃-Z membranes at 60 °C. The ionic conductance of CdPS₃-Al membrane at 30% RH was below the minimum detection limit of the device and thus not included.

Fig. R2 XRD patterns of CdPS₃-Al membrane at different RHs.

2. The polarization plot measured by the authors was swept towards only one direction (Sup Fig.15). What is the performance of the membrane under the cyclic voltammetry? Is the rapid transport of cations reversible?

Reply: We thank the reviewer very much for the insightful comments.

According to the reviewer's suggestions, we measured the polarization plots of CdPS₃-Z membranes by chronoamperometry at different voltage directions at 30 °C and 98% RH. As shown in Fig. R3, a significant current was observed upon a forward voltage activation, it then decreased, and finally reached a steady state with a reduced amplitude over time. When an opposite voltage was applied, reverse current was observed and it showed similar time-dependent behavior with forward current. Moreover, the transference numbers were nearly identical for both cases ($t_{ion} > 0.99$).

Fig. R3 Polarization plot measured by chronoamperometry at different voltage directions at 30 °C and 98% RH.

We further investigated the electrochemical performance of the membrane under cyclic voltammetry at 30 °C and 98% RH. As shown in Fig. R4, the presence of polarization at the scan rate of 0.03 V s⁻¹ from -0.6 V to 0.6 V is indicative of ion transport rather than electron transport. Importantly, the cyclic voltammetry curves show symmetric nature, with nearly identical area for the charging and discharging curves, indicating that the transport of cations is reversible.

Fig. R4 Cyclic voltammetry curves of CdPS₃-K membrane at 30 °C and 98% RH.

We have included the polarization plot and related discussions in the revised manuscript.

3. The fast permeation of cations is a very interesting phenomenon. But the authors need to comment on the stability of the cation-exchanged membrane. Do the cations get replaced while using the membranes in different conditions?

Reply: We thank the reviewer very much for the insightful comments.

We have evaluated the stability of the cation-exchanged CdPS₃-Z membranes by immersing them in 0.5 M LiCl solutions. XPS measurements show that K⁺, Na⁺, Ca²⁺ and Mg²⁺ can be entirely replaced by Li⁺, while Al³⁺ cannot be completely replaced (Fig. R5). However, we would like to emphasize that the main potential applications of these membranes are solid electrolytes in batteries or capacitors, where only one kind

of ions are involved. Thus, even if the interlayer ions can be re-exchanged by other ions, it would have no influence on their uses for batteries or capacitors.

Fig. R5 a-e, The quantified atomic ratio of elements of CdPS₃-K (a), CdPS₃-Na (b), CdPS₃-Ca (c), CdPS₃-Mg (d), and CdPS₃-Al (e) membranes immersed in 0.5 M LiCl for over 48 h.

To further demonstrate the potential application of our membranes, we fabricated all-2D solid-state micro-supercapacitors (MSCs) by using CdPS₃-Li membranes as solid ionic conductor and MXene membranes as electrodes, respectively (Fig. R6a-d). Notably, the stable voltage window of devices can reach 1.2 V at RT and 98% RH (Fig. R6e-h), which is comparable to the electrochemical thermodynamic stability window of water and much higher than those of the MXene/gels solid-state MSCs. Benefiting from the broadened voltage window, high areal and volumetric energy density of 4.3 $\mu\text{Wh cm}^{-2}$ and 28.8 mWh cm^{-3} are achieved at the power density of 60 $\mu\text{W cm}^{-2}$ and 400 mW cm^{-3} , respectively (Fig. R6h). Such performances are superior to those of the reported MXene/gels solid MSCs [*Energy & Environ. Sci.* 9, 2847-2854, 2016; *Nat. Commun.* 10, 1795, 2019; *Nano Energy* 69, 104431, 2020; *Adv. Funct. Mater.* 32, 2109593, 2021; *J. Mater. Chem. A* 5, 19639-19648, 2017; *Chinese Chem. Lett.* 27, 1586-1591, 2016; *Small* 14, 1801203, 2018]. Furthermore, our devices can be operated in a wide range of -10 to 50 °C and repeatedly bent without big performance change (Fig. R6i). The two serially connected MXene/CdPS₃-Li MSCs are capable of powering a light-emitting diode (LED) (Fig. R6j). These results demonstrate the great promise of our CdPS₃-Z membranes for practical applications.

Fig. R6 a, Schematic illustration of the fabrication of MXene/CdPS₃-Li solid-state MSCs. b-d, Cross-sectional SEM image and the corresponding EDS mappings of the interface region between CdPS₃-Li (SICs) and MXene (electrodes) membranes. Scale bar, 2 μm . e, f, CV curves at different potential windows (e) and scan rates (f). g, Galvanostatic charge and discharge profiles obtained at current density varying from 0.07 to 0.7 mA cm^{-2} . h, Comparison of voltage and volumetric energy density of MXene/CdPS₃-Li solid-state MSCs with the reported solid-state MXene/gels MSCs. i, CV curves at different operating temperatures. j, Photograph of a lit LED powered by two tandem MXene/CdPS₃-Li MSCs under bending.

We have expanded the discussion on the ion exchange order in the revised manuscript.

Reviewer #2 (Remarks to the Author):

In the paper the authors have described a super-ionic conductor i.e. CdPS₃ based membranes for the applications in wide temperature range from -40 to 90 degree Celsius. The authors have observed that the ionic conductivity of these membranes is

highest till reported. The main point of the paper is the highest membrane conductivity at temperatures below zero degree Celsius. The authors observe that for their membranes, the ionic conductivity remains more or less the same (line 115) up to -40 degree Celsius. While the observations are very interesting from a fundamental point of view, there are many grey areas, which needs to be addressed before making any decision on the publication.

Reply: We thank the reviewer very much for the positive comments and insightful suggestions that have helped us greatly improve the quality of our manuscript.

Using various characterizations, the authors have found that upon intercalation, the interlayer spaces of these membranes are occupied by cations in large numbers exceeding previous reports (line 154) and the ions exhibit mobilities much larger than the bulk (line 174). The authors have given a model based on concerted ion migration for explaining the high values of conductivities (line 144-163). It basically states that the negatively charged Cd vacancies attract lots of cations to the interlayer spaces and the large number of cations move together and have a low ion migration energy barrier that give rise to high mobility. This point needs careful consideration. The cartoon provided in figure 4a suggests one-dimensional transport, however, two-dimensional ion-migration picture should lead to a lower mobility. It is not clear as how the hopping rates are enhanced in this picture.

Reply: We thank the reviewer very much for the kind reminder.

We used electrochemical impedance spectroscopy (EIS) to measure the ionic conductivity within the CdPS₃-Z membranes. During testing, the ions within the membrane primarily move back and forth between the two electrodes in response to the applied alternating voltage. The schematic diagram in the original manuscript illustrates the ion transport along the direction of the two electrodes. The ion transport within a two-dimensional nanochannel appears similar to multiple one-dimensional ion transport, even though it is technically not the same. To clearly illustrate the concerted movement of ions within 2D nanochannels, we have revised the cartoon in the revised manuscript to avoid misunderstandings (Fig. R7).

As stated in the main text, charge-free pristine CdPS₃ and negatively charged Cd vacancies construct high-energy and low-energy sites for cation occupancy, respectively, in the 2D CdPS₃-Z nanochannels (Fig. R7a). The electrostatic attraction causes some of the cations to gather around the low-energy sites, while some cations occupy high-energy sites due to the high packing density of cations within the nanochannels (Fig. R7b). Moreover, the hydration shells prevent the cations from trapping into the vacancies. During ion transport, the cation located at the high-energy sites migrate downhill, which cancels out a part of the energy barrier felt by other uphill-climbing cations (Fig. R7c). This results in a low migration energy barrier and consequently high mobility of the cations. Conversely, if there are only low-density or limited cations in the nanochannels, they prefer occupying the low-energy sites due to electrostatic attraction and thus a high energy barrier must be overcome during cation transport. As a result, the effective hopping rate of the cations decreases.

Fig. R7 a, Schematic illustration of the structure of CdPS₃-Z membranes with high-energy sites (around charge-free pristine CdPS₃) and low-energy sites (around negatively charged Cd vacancies) for cation occupancy. **b**, **c** The distribution of cations within the nanochannels of CdPS₃-Z membranes along the direction of applied voltage (b) and the corresponding energy landscape (c) for the concerted migration.

Further, Al³⁺ mobility is similar to bulk, and there is no reason to think that there is any hydration picture that needs to be invoked either. The interlayer space do not show any trend with cations, indicating that hydration radius is not a good parameter to discuss. The authors should have tried aqueous salt solution-based conductivity experiments to understand the nature of the walls as all of the experiments were done at 98% humidity. In addition, the authors also discuss the stronger interaction of water with cations to explain the decreased freezing point of water. The stronger interaction would lead to a hydration dominated transport, which the authors are unable to observe. The contact angle is pointing to a different picture for Al³⁺, with a contact angle of 63 degrees. The less hydrophilicity would suggest a reduced surface charge, leading to less electrostatic interactions with the ions and hence recovering to the bulk mobility for Al³⁺. This also suggests that electrostatic interactions can fully explain the experimental result. According to the Raman characterization by the authors, below the freezing temperature, the structure of water inside the interlayer spaces is not similar to bulk ice (supplementary fig 18). It has a lower number of strong hydrogen bonds and is not in the completely ordered state as that of bulk ice. Also, there are just two layers of water molecules inside the interlayer space and most of them are interacting with cations, leaving no place for hydrogen bonds between water molecules. Since there is no change in the state of water, mobility remains fairly constant.

Reply: We thank the reviewer very much for the kind reminder.

The hydration of cations in the 2D nanochannels have been widely reported [e.g., *Science* 358, 511-513, 2017; *Nat. Nanotechnol.* 12, 546-550, 2017]. Normally, the ion transport behavior strongly depends on the hydration radius due to the steric effect. According to the XRD data and the corresponding interlayer distance, all ions are

hydrated inside the nanochannels in our experiments at 98% RH (Fig. 1 d, e). In addition, taking into account the high-density cations confined within the 2D nanochannels, the coordination of water molecules with cations would destroy the H-bond interactions between water molecules, enabling the survival of non-H-bonded water molecules, and thus the decrease of freezing point, which are confirmed by Raman measurements (Fig. 4 e, f). Based on the different hydration radius and the different electrostatic interactions with the channel walls for different cations, it is expected to observe significantly different ion conductivities for different cations as reported previously [e.g., *Science* 358, 511-513, 2017; *Nat. Rev. Mater.* 5, 229-252, 2020; *Chem. Mater.* 34, 881-898, 2022]. In contrast, in the present study, we observed that all the membranes exhibit unexpectedly similar superhigh ion conductivities for monovalent (K^+ , Na^+ , Li^+) and multivalent ions (Ca^{2+} , Mg^{2+} , Al^{3+}), ~ 0.01 to 0.8 S cm^{-1} in the temperature range of $-30 - 90 \text{ }^\circ\text{C}$, which are one to two orders of magnitude higher than those of the corresponding best solid ion conductors. This is the main finding of our manuscript.

The main potential applications of CdPS₃-Z membranes are solid-state ion electrolytes rather than ion-sieving films. Typically, impedance spectroscopy is employed to study the ion transport in quasi-/solid-state ion electrolytes at varied temperature and humidity conditions. Using impedance spectroscopy can avoid any potential ion sieving that may occur when ions attempt to transport from the bulk solution to the nanochannels, which is an issue for solution-based conductivity experiments. Thus, the intrinsic ion transport within the nanochannels can be identified.

We agree with the reviewer that the increase in the contact angle suggests a reduced surface charge. To understand the origin, we tested the contact angle of CdPS₃-Al membrane after it was taken out of solution and left under ambient conditions for varying durations. As shown in Fig. R8a, the contact angle is around 30° at the beginning, and it increases with drying time and reaches $\sim 60^\circ$ after one hour. Moreover, the sample shows a distinct sharp XRD peak at 7.0° at the beginning, which corresponds to an interlayer distance $d = 1.26 \text{ nm}$. However, this XRD peak gradually decreases along with an increase of the XRD peak at 13.2° ($d = 0.67 \text{ nm}$) with extending drying time. After being left in ambient conditions for 5 days, the films displayed a weak and broad peak at 7.9° ($d = 1.11 \text{ nm}$) and a distinct peak at 13.2° ($d = 0.67 \text{ nm}$) (Fig. R8b). The significantly decreased interlayer distance indicates that the vacancies on the CdPS₃ nanosheets are re-occupied by Al^{3+} caused by the strong electrostatic attraction when the hydration shells are removed after drying, which results in the decrease in the number of surface charges.

Fig. R8 a, b Evolution of contact angle (a) and XRD pattern (b) of CdPS₃-Al membrane with drying time under ambient conditions.

We also studied the contact angle change of other membranes beyond CdPS₃-Al with drying time. Different from CdPS₃-Al membrane, both the contact angle and interlayer distance exhibit only minor differences for these membranes under high and low RHs (Fig. R9, Fig. 1d, Fig. S16). It is needed to point out that the ion transport experiments presented in our manuscript were all carried out at 98% RH. Thus, it is more reasonable to analyze the ion transport mechanism based on the structure and properties of the membranes under 98% RH, where cations remain in the nanochannels for all CdPS₃-Z membranes as confirmed by XRD data (Fig. 1d). In this case, the number of vacancies and the surface charge density are the same for all membranes. Compared to Li⁺, K⁺, Na⁺, Ca²⁺ and Mg²⁺, Al³⁺ has more charges and larger hydrated diameter, resulting in larger electrostatic attractions with the channel walls and stronger steric effect. Therefore, it shows lower mobility than those of other cations. For Ca²⁺ and Mg²⁺, the latter shows lower mobility than the former even though they have the same charges, indicating the hydrated diameter of the cations also plays an important role in their transport behavior.

Fig. R9 Comparison of the contact angle of CdPS₃-K, CdPS₃-Na, CdPS₃-Ca, CdPS₃-Mg, and CdPS₃-Al membranes under high and low RHs.

Moreover, it is worth noting that the mobility of Al³⁺ in angstrom-scale charge-free

slits is ~10 times lower than its bulk mobility due to the steric effect [*Science* 358, 511, 2017]. The electrostatic interactions indeed affect the ion transport in CdPS₃-Z membranes but it could not account for the higher mobilities of the cations confined in 2D nanochannels than in bulk solutions. As shown in our manuscript, we suggest that this unique ion transport behavior is due to the high packing density of cations within the nanochannels, which imparts strong repulsive Coulomb interaction among adjacent cations to overcome the electrostatic attractions from vacancies, and consequently promotes concerted movements of multiple ions.

1. The percentage of cations is calculated from ICP-AES, and the absence of chlorine is confirmed through EDAX. Although XPS has been done, it is not used to calculate the percentage of ions or absence of chlorine in the membrane. A survey spectra of XPS with depth analysis is also missing which would have been a better proof for the distribution and estimating the percentage of ions intercalated.

Reply: We thank the reviewer very much for the valuable suggestions.

We have conducted XPS tests to further analyze the percentage and distribution of different elements in the CdPS₃-Z membranes. To evaluate the uniformity of the membranes, they were etched by argon ions for different time. As shown in Fig. R10, the survey spectra of XPS with depth analysis indicates that no chlorine and lithium were detected and the atomic ratio of Cd, P, S and Z remained nearly unchanged at different depth along the thickness direction of the membrane. The surface tended to adsorb some substances beyond CdPS₃-Z when exposed to ambient conditions, which led to the slight difference in the chemical composition between the surface and the bulk. Although XPS is a surface-sensitive spectroscopic technique that can identify the elements in a material, obtaining absolute quantification is challenging since it requires the use of certified (or independently verified) standard samples. Generally, quantitative accuracy depends on many factors including signal-to-noise ratio, peak intensity, accuracy of relative sensitivity factors, correction for electron transmission function, surface volume homogeneity, correction for energy dependence of electron mean free path, and degree of sample degradation. Under optimal conditions, the quantitative accuracy of the atomic percent (at%) values calculated from the major XPS peaks is 90-95%. The quantitative accuracy for the weaker XPS signals, which have peak intensities 10-20% of the strongest signal, is 60-80% of the true value, and depends on the efforts used to improve the signal-to-noise ratio (for example by signal averaging). Considering the above influence factors, the obtained atomic percentages from XPS are comparable to the results obtained from ICP-AES tests.

Fig. R10 a,e,i,m,q, The calculated atomic ratio of different elements in CdPS₃-Z membranes based on survey of XPS spectra. **b-d**, K 2p, Cl 2p, Li 1s, **f-h**, Na 1s, Cl 2p, Li 1s, **(j-l)** Ca 2p, Cl 2p, Li 1s, **n-p**, Mg 2p, Cl 2p, Li 1s, and **r-t**, Al 2p, Cl 2p, Li 1s spectra of CdPS₃-K, CdPS₃-Na, CdPS₃-Ca, CdPS₃-Mg, and CdPS₃-Al membranes, respectively, after etching for different time.

We have added the above XPS results in the revised manuscript.

2. The large negative charge of the layers would make the layers to be very unstable, especially in water related applications. The large negative charge of the layers is a concern for fouling also. Did the authors tried any experiment to test the stability of these membranes in water?

Reply: We thank the reviewer very much for the insightful comment.

If there were no counter cations inside the nanochannels, the membranes would be very unstable in water, similar to graphene oxide membranes, because of the electrostatic repulsion between negatively charged neighboring nanosheets. However, in our membranes, the large number of cations in the nanochannels can attract the

negatively charged nanosheets together and consequently enables the membranes good stability in water. We have followed the reviewer's suggestion to test the stability of our membranes in water. As shown in Fig. R11, all the membranes except for CdPS₃-Li have no obvious change after immersing in water for 48 h. The CdPS₃-Li tends to swell and gets damaged when immersed in DI water (Fig. R12), which is reasonable to expect because Li⁺ has a small radius, a single electric charge and a highly hydrated shell and thus it has weaker electrostatic interaction with the negatively charged nanosheets compared with other cations. Due to the same reason, lithium salts have been widely used as intercalants for exfoliating layered materials. However, the incorporation of Li-containing salts in the water has resulted in a significant improvement in the stability of the CdPS₃-Li membrane. For example, when LiCl was added to the water, the CdPS₃-Li membrane can maintain its integrity without obvious change in 48 h (Fig. R13). The main applications of these membranes are supercapacitors or batteries, where the electrolyte containing the same kind of ions are used. Therefore, there should be no stability and fouling problems for such applications.

For example, we fabricated all-2D solid-state micro-supercapacitors (MSCs) by using CdPS₃-Li membranes as solid ionic conductor and MXene membranes as electrodes, respectively (Fig. R6a-d). Notably, the stable voltage window of devices can reach 1.2 V at RT and 98% RH (Fig. R6e-h), which is comparable to the electrochemical thermodynamic stability window of water and much higher than those of the MXene/gels solid-state MSCs. Benefiting from the broadened voltage window, high areal and volumetric energy density of 4.3 $\mu\text{Wh cm}^{-2}$ and 28.8 mWh cm^{-3} are achieved at the power density of 60 $\mu\text{W cm}^{-2}$ and 400 mW cm^{-3} , respectively (Fig. R6h). Such performances are superior to those of the reported MXene/gels solid MSCs [*Energy & Environ. Sci.* 9, 2847-2854, 2016; *Nat. Commun.* 10, 1795, 2019; *Nano Energy*, 69, 104431, 2020; *Adv. Funct. Mater.* 32, 2109593, 2021; *J. Mater. Chem. A* 5, 19639-19648, 2017; *Chinese Chem. Lett.* 27, 1586-1591, 2016; *Small*, 14, 1801203, 2018]. Furthermore, our devices can be operated in a wide range of -10 to 50 °C and repeatedly bent without big performance change (Fig. R6i). The two serially connected MXene/CdPS₃-Li MSCs are capable of powering a light-emitting diode (LED) (Fig. R6j). These results demonstrate the great promise of our CdPS₃-Z membranes for practical applications.

Fig. R11 The stability of CdPS₃-K, CdPS₃-Na, CdPS₃-Ca, CdPS₃-Mg and CdPS₃-Al membranes in water. Scale bar: 1 cm.

Fig. R12 The stability of CdPS₃-Li membrane in water. Scale bar: 1 cm.

Fig. R13 The stability of CdPS₃-Li membrane in 0.5 M LiCl. Scale bar: 1 cm.

We have added the stability results in the revised manuscript.

3. Estimation of mobility can be erroneous as it has been estimated from the conductivity.

Reply: We thank the reviewer very much for the insightful comment.

While it is true that there might be potential sources of error when estimating mobility from conductivity, it is a widely accepted and commonly used method in many studies. For example, Huang's group used I - V curves to obtain ion conductivity in clay-based 2D nanofluidic channels, and based on the conductivity values, both the surface charge density and ion mobility were estimated [*Nat. Commun.* 6, 7602, 2015]. Geim and coworkers estimated the ion mobilities of various ions in ultraclean 2D slit nanofluidic channels from the ion conductivities [*Science* 358, 511-513, 2017]. They assumed that the ion concentration in the channels was equivalent to the bulk concentration and the calculations were made using the same equation as ours. This method has also been widely used to calculate the ion mobility of solid-state ion conductors to further obtain the ion diffusion rate in the literatures [e.g., *Chem. Mater.* 34, 881-898, 2022; *Energy Sci. Engineer.* 10, 1643-1671, 2021].

4. The estimation of conductivity is geometry dependent and the authors need to provide the actual parameters that are used for the estimation.

Reply: We thank the reviewer very much for the kind suggestion.

The samples used in our study had a rectangular shape with dimensions of 15 mm × 5 mm (length × width), and their thickness varied slightly between samples. We measured the thickness of each sample using cross-sectional SEM imaging.

We would like to point out that the conductance of a sample is geometry dependent, as it is proportional to the length of the sample and inversely related to the cross-

sectional area. However, the influence of geometry has been eliminated in the calculation of conductivity (S cm^{-1}):

$$\sigma = \frac{L}{R \cdot S}$$

where L (cm) is the length of membrane, R (Ω) is the resistance calculated from Nyquist plots, and S (cm^2) is the cross-sectional area of the membrane.

We have included the information about the geometry of our sample in the revised manuscript.

5. It is not clear how the authors estimated transference number from the chronoamperometry data. It needs to be explained at least in the supplementary text.

Reply: We thank the reviewer very much for the valuable suggestion.

In our study, we used chronoamperometry to differentiate the electronic and ionic conductance in the membranes. This electrochemical technique measures the current as a function of time by applying a constant potential to the electrodes. An inert Pt electrode was used to block ionic transport. After the device was subjected to a direct voltage, ions accumulated at the interface between the tested membrane and Pt electrode, creating a significant internal inverse electric field that inhibited ion transport (Fig. R14a). As ions were blocked, the current was primarily carried by electrons, allowing us to distinguish between the ionic and electronic contributions. Taking CdPS₃-K membrane as an example, as shown in Fig. R14b, at the initial stage of measurement, the current (i_T) was derived from both ionic and electronic current. However, in the static state, the ionic conductance was blocked and only the electronic current was measured (i_e). Thus, the ionic transference number (t_{ion}) was calculated as follows:

$$t_{ion} = \frac{i_T - i_e}{i_T}$$

In our experiments, the extracted t_{ion} is ~ 0.99 . This result suggests that the electronic conduction contribution is negligible and the CdPS₃-Z membrane is a ion conductor.

Fig. R14 a, Schematic illustration of the chronoamperometry method with inert electrodes employed. **b**, Evolution of current by chronoamperometry ($V = 1$ V) for CdPS₃-K membranes at 30 °C and 98% RH.

We have added the above information in the revised manuscript.

6. The activation energy plot clearly shows two slopes. At low temperatures, the energy barrier for ion migration is slightly higher. The activation energy mentioned for the intercalants do not show any trend or the variations are within the error bar. However, the mobility clearly shows a decreasing trend with increase in valence, which is similar to the trend in contact angle, at least for the end points.

Reply: We thank the reviewer very much for the kind comment.

Both the charges and size can significantly affect the mobility and conductivity of ions in the nanochannels. Typically, the conductivities of multivalent ions are at least one order of magnitude smaller than those of monovalent analogues for the known solid ion conductors [*Science* 358, 511-513, 2017; *Nat. Rev. Mater.* 5, 229-252, 2020; *Chem. Mater.* 34, 881-898, 2022; *J. Am. Chem. Soc.* 139, 13260-13263, 2017]. It has also been reported that the mobility of Al^{3+} in angstrom-scale charge-free slits is substantially decreased over 10 times compared to those of K^+ , Na^+ , and Li^+ due to steric effect [*Science* 358, 511-513, 2017]. However, our membranes only show a slight decrease in ion conductivity and mobility as hydration diameter and charges of cations increase. This is consistent with the similar activation energy values for $\text{CdPS}_3\text{-Z}$ membranes. As mentioned in the above response, all the $\text{CdPS}_3\text{-Z}$ membranes show similar contact angles under the same ionic conductivity measurement conditions (98% RH). The larger contact angle for $\text{CdPS}_3\text{-Al}$ membranes presented in our original manuscript is due to the removal of water from the nanochannels after drying.

7. There is no mention of how many samples showed similar behavior and what is the percentage of samples that showed the same trend. Error bars are missing from the plots.

Reply: We thank the reviewer very much for the kind reminder.

In our experiments, more than three samples were measured for each type of membranes and all of them showed the same behavior. In the original manuscript, the energy barriers were actually the average energy barriers, which were calculated based on the average of ion conductivities, hence no error bars were included. In the revised manuscript, we calculated the energy barrier based on each ion conductivity, and included error bars accordingly (Fig. R15).

Fig. R15 E_a at normal- and low-temperature stages for $\text{CdPS}_3\text{-Z}$ membranes. Error bars represent s.d.

8. The contact angle of CdPS₃-Li is lower than CdPS₃-Na. Ideally, Li should be more hydrophilic than Na. Is the observed contact angle within the error bar?

Reply: We thank the reviewer very much for the insightful comment.

If the interlayer cations are the only difference between CdPS₃-Na and CdPS₃-Li membranes, the contact angle of CdPS₃-Na membrane should be lower than that of CdPS₃-Li membrane, due to the generally expected lower hydrophilicity of Na⁺. However, different from CdPS₃-Li membranes, we noticed some humps on the surface of CdPS₃-Na membrane. Given that the surface roughness can affect the contact angle measurement by altering the apparent contact area between water and the surface, we hypothesized that the increased surface roughness of CdPS₃-Na contributes to the enhanced hydrophilicity to a similar degree as the CdPS₃-Li membrane.

9. The studies are done at 98% RH. The stability of the membranes if dipped in water, is not discussed at all.

Reply: We thank the reviewer very much for the valuable suggestions.

As we previously stated in our response to the second comment, all CdPS₃-Z membranes except for CdPS₃-Li have good stability in water. However, with addition of Li-containing salts, CdPS₃-Li has shown significantly improved stability in the solution and no noticeable change was observed within 48 h. Given that the membranes are typically used in salt-rich environments, such as batteries or supercapacitors, they should be stable enough for practical applications.

10. In line 80, it is mentioned that abundant cations make the membrane hydrophilic, so why is aluminum less hydrophilic than the rest? How is the charge playing a role here in hydrophilicity? Contact angle clearly shows a large variation.

Reply: We thank the reviewer very much for the insightful comment.

In our study, the CdPS₃-Z membranes contain negatively charged Cd vacancies and interlayer cations, which can interact with polar water molecules through electrostatic interactions and therefore play an important role in determining the hydrophilicity of the membrane.

Several factors are likely to affect the contact angle of CdPS₃-Z membranes. First, the surface chemistry can significantly affect the contact angle. For all CdPS₃-Z membranes, the main difference in surface chemistry is likely to be the type of cations present in the nanochannels, as the basic planes have the same composition and structure. For the CdPS₃-Li membrane, the excellent hydrophilicity can be attributed to the abundance of Li⁺ in the interlayer, compared to multivalent ions. Meanwhile, Li⁺ ions have a smaller crystal radius than other monovalent ions [*J. Phys. Chem.* 63, 1381, 1959], which results in stronger interactions with water. This is further supported by the facts that lithium salts have been widely used as intercalants for exfoliating layered materials such as CdPS₃ and vermiculite crystals, and the CdPS₃-Li membrane is unstable in DI water because of the excessive absorption of water caused by Li⁺. Second, the surface roughness of the surface can affect the contact angle measurement by altering the apparent contact area between water and the surface. For instance, we observed some bumps on the surface of CdPS₃-Na membrane. The increased surface

roughness enabled the membrane enhanced hydrophilicity in comparison to other membranes with similar surface chemistry but smooth surfaces. Third, while other factors such as testing time and droplet size may fall within the margin of human and machine error, they can still have an influence on the results. Therefore, the contact angle shows some variations for different membranes.

As we mentioned in the response to the first comment, the CdPS₃-Al membrane has similar contact angle with other membranes at high hydrated state. As shown in Fig. R8a, the contact angle is around 30° for the hydrated CdPS₃-Al membrane, and it increases with drying time and reaches ~60° after 24 hours. Moreover, the sample shows a distinct sharp XRD peak at 7.0° at the beginning, which corresponds to an interlayer distance $d = 1.26$ nm. However, this XRD peak gradually decreases along with an increase of the XRD peak at 13.2° ($d = 0.67$ nm) with extending drying time. After being left in ambient conditions for 5 days, the films displayed a weak and broad peak at 7.9° ($d = 1.11$ nm) and a distinct peak at 13.2° ($d = 0.67$ nm) (Fig. R8b). This phenomenon is similar to MoS₂ membrane, which maintains a 1.2 nm interlayer distance at the fully hydrated state, but undergoes a reduction in the interlayer distance to 0.62 nm after water removal from the interlayer space and subsequent nanosheet restacking [*Nano Lett.* 17, 7289, 2012]. The above results indicate that the vacancies on the CdPS₃ nanosheets are re-occupied by Al³⁺ when the hydration shells are removed after drying, which results in the decrease in the number of surface charges and consequently the increase of contact angle. The easy dehydration of CdPS₃-Al membrane might be due to the larger electrostatic attraction between Al³⁺ and the negatively charge nanosheets.

In the revised manuscript, we have expanded the discussion of the surface charge and contact angle relationship with a focus on the CdPS₃-Al membrane.

11. For XRD, in line 88, it is mentioned that there is sheet restacking at low relative humidity, due to which there is a bulk peak (13.2 degrees) at low RH; can this be plausibly due to incomplete exchange? Apart from this, the extra peaks above 15 degrees at low RH need to be properly accounted for.

Reply: We thank the reviewer very much for the insightful comment.

If the XRD peak at 2θ of ~13.2° was caused by the unexfoliated flakes, it would not disappear at high RH. The humidity alone cannot cause the unexfoliated flakes to become exfoliated without any additional measures or external intervention. Furthermore, extensive AFM measurements demonstrate that the CdPS₃ nanosheets are predominantly monolayers, with thickness of ~1 nm (Supplementary Fig.1). Inductively coupled plasma-atomic emission spectroscopy (ICP-AES) is an analytical technique that has been widely used to detect and quantify trace elements in a sample. In our study, we utilized ICP-AES to determine the composition of the samples. The results show that there was no Li⁺ element left after the exchange process. In addition, the survey spectra of XPS with depth analysis also indicates that no lithium was detected (Fig. R10q). As for the presence of a similar peak in the bulk crystals, the ~13.2° peak observed in CdPS₃-Z membrane is attributed to the restacking of the nanosheets at low RH. In our previous study [*Science* 370, 596-600, 2020], the loss of

water in CdPS₃-Li membrane at low RH have been confirmed which results in the restacking and decrease in the interlayer distance.

The XRD peak above 15° for the CdPS₃-Z membrane corresponds to the second-order diffraction of the (001) planes. For example, for the CdPS₃-K membrane, the first order reflection from (001) planes appears as a peak at 2θ of $\sim 9.5^\circ$. By using the equation $2d_{001} \sin \theta = \lambda$, we can get a d_{001} value of 9.3 Å. Using equation $2d_{001} \sin \theta' = 2\lambda$, we determined that the second-order reflection from (001) occurs at 19.1° , which corresponds to the peak above 15° in the XRD pattern of CdPS₃-K (Fig. S16). This indicates that the peak above 15° is a result of the second-order reflection from the same planes that gave rise to the first peak. This relationship between the first peak and the peak above 15° applies to all the membranes except CdPS₃-Al membrane shown in Fig. S16.

We have added the origin of the XRD peak above 15° in the revised manuscript.

12. On line 89, XRD peak at 13.5 degree mentioned to disappear, however, in the figure it is still visible.

Reply: We thank the reviewer very much for the insightful comment.

In our previous article [*Science* 370, 596, 2020], we did a thorough investigation on the XRD peak evolution of Cd_{0.85}PS₃Li_{0.15}H_{0.15} and Cd_{0.85}PS₃Li_{0.3} (CdPS₃-Li) along with RH. Here, we used the CdPS₃-K membrane as an example to explain the evolution of XRD peaks observed in the present study. At low RH, the presence of some cations in the reassembled membrane results in a peak at 2θ of $\sim 9.5^\circ$ in the XRD spectrum. The nanosheets restacking in certain regions leads to the appearance of a peak at 2θ of $\sim 13.2^\circ$, similar to that of CdPS₃ crystal. As mentioned in the above reply, the peak at 2θ of 19.1° is the second-order reflection from (001) planes. At high RH, continued incorporation of water molecules results in the formation of a bilayer water network, which leads to an increase in the interlayer distance and the disappearance of the original peak at 13.2° . Meanwhile, the first-order reflection from (001) planes shifts to 6.8° , while the corresponding second-order reflection shifts to 13.6° . Therefore, at low RH, the peak observed at around 13.2° is due to the restacking of nanosheets in the regions without hydrated cations in the nanochannels (Supplementary Fig. S16). While the peak observed at around 13.6° at high RH originates from the second-order reflection from (001) planes (Fig. 1d).

13. y-axis scale of Supplementary Fig 15 a) can be represented better, e.g. from 0-20 so that the variation can be more evident.

Reply: According to the reviewer's kind suggestion, we have optimized the figure in the revised manuscript (Fig. R16).

Fig. R16 Polarization plots measured by chronoamperometry ($V = 1$ V) at 30 °C and 98% RH. Inset, zoom-in view of polarization plots at the beginning.

14. Line 119-Is -30 deg a typo? Is it meant to be 30 deg? Otherwise -30 deg is not in the Fig. Lowest conductivity temperatures are different for different cations. Is there any reason?

Reply: We thank the reviewer very much for the kind reminder.

We have checked Line 119 and found no typo. Achieving practical conductivity levels (~ 0.01 S cm^{-1}) for quasi-/solid-state electrolytes is challenging. Typically, high ion conductivity can only be attained at elevated temperatures, as the conductivity of most electrolytes rises with temperature [*Nat. Rev. Mater.* 5, 229-252, 2020; *Energy & Environ. Sci.* 11, 1945-1976, 2018]. In contrast, the ion conductivity in CdPS₃-Z membranes is quite high, which can achieve the practical conductivity level even at low temperatures. Specifically, CdPS₃-K, CdPS₃-Na and CdPS₃-Ca membranes exhibit practical conductivity at temperature above -30 °C, while CdPS₃-Na, CdPS₃-Mg and CdPS₃-Al membranes display practical conductivity at temperature above -20 °C (Fig. R17). We included the data tested at -20 °C in Fig. 2b since all the membranes can achieve practical conductivity above this temperature.

We have included the ion conductivities of CdPS₃-Z membranes at -30 °C into the Fig.2b in the revised manuscript.

We have also tried to test the ion conductivities of all the membranes at -40 °C. However, we were only able to test the CdPS₃-Al membrane down to -35 °C, as the resistance exceeded the detection limit at -40 °C. Thus, at -40 °C, the ion conductivities of all the membranes except for CdPS₃-Al membrane were given in Fig. 2a. As pointed out in our manuscript, the high-density cations destroy the H-bond interactions between water molecules confined within the 2D nanochannels, which enables the survival of non-H-bonded water molecules and thus high ion conductivities below the freezing point of water (0 °C). Notably, the CdPS₃-Al membrane has lower cation density than other membranes. This is very likely to be an important reason for the rapid decline of its ion conductivity below 0 °C and consequently the higher temperature corresponding to the lowest conductivity that could be tested.

Fig. R17 Conductivities of CdPS₃-Z membranes measured at 90, 30, -20 and -30 °C and 98% RH.

15. Some references can be given for lines 135-138.

Reply: According to the reviewer's kind suggestion, we have included appropriate references to support the content discussed in lines 135-138 in the revised manuscript.

Reviewer #3 (Remarks to the Author):

This work investigates the conductivity of 2d materials with high density of ion intercalation with the aim to develop highly conductive solid ionic conductors. They report a class of versatile superionic conductors, monolayer CdPS₃ nanosheets-based membranes intercalated with diverse cations with a high density. The high conductivity is even demonstrated under a temperature as low as -30 degree C. Their finding benefits the design of solid ionic conductors. However, I have following questions and comments.

(1) The authors use an immersion method to replace Li⁺ with other ions inside the 2d layer. It is known that Li⁺ has lower hydration radius and thus lower dehydration energy than multivalent ions. It is counter-intuitive that those multivalent ions can replace Li ions inside the 2D channels. As shown in the paper of [Nature 550, 380-383 (2017)], the weak cation- π interaction of potassium ions can prevent the intercalation of ions with large hydration radius. This work claims the surface charge density as high as 2 e/nm². The electrostatic interaction between 2D layers and cations should be much higher and more stable, rendering it difficult for lithium ions to be replaced by multivalent ions. Thus, it is necessary to analyze the energy barrier for the ion replacement.

Reply: We thank the reviewer very much for the insightful comments.

In the paper [Nature 550, 380-383 (2017)], the authors reported ion sieving in graphene oxide (GO) membranes via cationic control of interlayer spacing. They found that the fixing of the interlayer distances is mainly due to the interaction between the hydrated cations and aromatic rings (cation- π interactions) on the GO sheets, as well as the interaction between the hydrated cations and the oxidized groups on the GO sheets.

Molecular orbital analysis reveals the coupling between lone pair of electrons of the oxygen atoms in the oxygen functional groups and the delocalized π states of the aromatic ring structure in GO and the empty orbitals of the cation. Notably, the cation- π interaction also arises from the electrostatic attraction between the positive charge of the cation and the electron-rich π -cloud of the aromatic ring. In fact, this type of interaction is relatively strong and it is difficult to break because cation- π interaction involves a continuous region of electron density (the π -cloud). As a result, cations are tightly bound to GO through cation- π interactions and cannot be easily exchanged by other cations due to steric exclusion and ion sieving.

In contrast, the charges present on the surface of the nanochannel walls in CdPS₃-Z materials make the interlayer ions loosely bound to the surface through electrostatic interactions, forming a material similar to ion exchangeable materials such as vermiculite. In vermiculite or the CdPS₃ based membrane, the electrostatic interaction is limited to specific sites where the negative charges are located. Once the cations in the interlayer spacing deviate from the sites due to thermal vibration or other factors, the electrostatic interactions between ions and exchangeable sites weaken, which allows for ion exchange to occur. According to density functional theory calculations, our previous work revealed that protons can easily detach from the nanosheet and then attach to water clusters, with an energy change of -1.199 eV [*Science* 370, 596-600, 2020]. Notably, this exothermic process provides evidence of a weak interaction between the cations and the vacancy. As a result, despite high surface charge density, cations in the interlayer spacing can still be exchanged driven by the concentration gradient, electrostatic attraction, and capillary-like pressure. There appears to be no obvious order of ion substitution in ion-exchange materials. Although the hydrated radius of multivalent cations is larger than that of monovalent cations, ion exchange can still occur for multivalent cations. Vermiculite, for instance, can replace interlayer hydrated Na⁺ with Cd²⁺ ions in solution, making it a popular adsorbent for multivalent ions in treating heavy metals in wastewater [*Environ. Sci. Pollut. Res.* 29, 79903-79919, 2022]. Nair and his colleagues successfully exchanged Li⁺ ions in the interlayer spacing with other cations such as K⁺, Ca²⁺, La³⁺, and Sn⁴⁺, and discovered that the wetting properties of vermiculite membranes can be regulated through this ion exchange process [*Nat. Commun.* 11, 1097, 2020]. When vermiculite is exfoliated, the original divalent Mg²⁺ in the interlayer can be replaced with monovalent Na⁺, which can then be exchanged with Li⁺ [*Nat. Commun.* 6, 7602, 2015]. Moreover, it has been also found when mica was immersed in aqueous solution, the interlayer K⁺ ions were readily exchanged with other cations present in the solution, for example H⁺, Cs⁺, Ba²⁺ [*J. Phys. Chem.* 61, 1408-1413, 1957] and Ca²⁺, Na⁺ [*Carbohydr. Polym.* 34, 146-151, 1970], depending on the concentration of each ion [*J. Phys. Chem.* 61, 1408-1413, 1957; *Carbohydr. Polym.* 83, 531-546, 1981]. Thus, unlike GO materials, the replacement order in ion exchangeable materials represented by vermiculite or mica appears to be less influenced by the size of hydrated ions.

We further studied the ion exchange in CdPS₃-Z membranes. As shown in Fig. R10, Li⁺ confined in the interlayer can be completely exchanged to K⁺, Na⁺, Ca²⁺, Mg²⁺, and Al³⁺. Meanwhile, K⁺, Na⁺, Ca²⁺ and Mg²⁺ in the interlayer can be completely exchanged

back to Li^+ ions but Al^{3+} cannot be completely exchanged back anymore (Fig. R5). Moreover, hydrated K^+ with the smallest hydration radius in the interlayer also can be exchanged to Na^+ , Ca^{2+} and Al^{3+} (Fig. R18). These results confirm the weak electrostatic interaction between cations in the interlayer and the negatively charged surface, enabling that multivalent ions with large hydration radius indeed can replace monovalent ions with small hydration radius inside the 2D channels. The weak electrostatic interaction is also an important reason for the ultrahigh ion conductivity observed in CdPS₃-Z films. Further theoretical investigations are required to reveal the underlying principles and mechanisms associated with the ion exchange in CdPS₃-Z membranes in the future.

Fig. R18 a-c, The quantified atomic ratio of elements in CdPS₃-K membranes measured by XPS spectra with depth analysis after immersed in 0.5 M NaCl (a), 0.5 M CaCl₂ (b), and 0.5 M AlCl₃ (c) for over 48 h. These results suggest that K^+ in the CdPS₃-K membrane was entirely replaced by the corresponding ions.

We have expanded the discussion on the ion exchange order in the revised manuscript.

(2) The authors claim the surface charge density of Cd_{0.85}PS₃Li_{0.3} is on the order of 2 e/nm² based on the composition analysis. Does surface zeta potential measurement support this argument?

Reply: We thank the reviewer very much for the valuable suggestion.

We tested the zeta potential of CdPS₃-Li nanosheets dispersion, which is about -51.8 mV (Fig. R19), indicating the nanosheets are highly negatively charged. Zeta potential is related to the surface charge density of a particle or surface, but the relationship between them is not straightforward. The zeta potential measured by a surface zeta potential is a result of the balance between the charges on the surface and the ions in the surrounding solution. In addition to the surface charge density, many other factors such as particle size, shape, composition, and the surrounding medium can also influence the zeta potential. Thus, it is difficult to identify the surface charge density from zeta potential. In our study, the surface charges were calculated based on the crystal structure and chemical composition analysis, where the contribution of ions in the surrounding solution to the overall surface charge was not taken into account, thus providing the real surface charge density of the sample.

Fig. R19 Zeta potential of CdPS₃-Li nanosheet dispersion.

We have added the zeta potential results in the revised manuscript.

(3) In fig.2, the energy barrier for ions movement inside the 2d channels can be extracted from the Arrhenius plots. Why does the sodium ion show the highest energy barrier in the high temperature range, and the Potassium ion show the second largest energy barrier in the low temperature range, as told from the slopes?

Reply: We thank the reviewer very much for the insightful comments.

Since the energy barrier was calculated from the slope of the fitted line of ion conductivity, a slight change of the ion conductivity will affect the value of the energy barrier. Actually, there is no big difference in the energy barrier for the movement of different ions in both the high temperature range and low temperature range (Fig. 4b). For example, Na⁺ and Li⁺ show the highest and lowest energy barrier in the normal temperature range, respectively, but the energy barrier difference is only 0.1 eV. The relatively larger energy barrier for Na⁺ might be due to the smallest interlayer distance of CdPS₃-Na membrane (Fig. 1e).

As mentioned in our manuscript, we suggested that the coordination of water molecules with cations destroys the H-bond interactions between water molecules, enabling the survival of non-H-bonded water molecules and consequently ensures a high ion conductivity below the freezing point of water. Variations in the number of cations and their capacity to bind with water molecules can result in differences in the ratio of different H-bonded water. For example, although Al³⁺ has a strong binding with water molecules, the number of trivalent Al³⁺ is significantly less than other ions. As a result, more water molecules freeze at low temperatures, leading to the largest E_a of CdPS₃-Al membrane at low temperature. In contrast, although K ions are present in larger numbers, their capacity to bind with water molecules is limited with the lowest hydration energy among all the studied ions [*J. Chem. Soc. Faraday Trans. 87*, 2995-2999, 1991]. This means that compared to CdPS₃-Li and CdPS₃-Na nanochannels, more water molecules will freeze in the nanochannel of CdPS₃-K as temperature decreases, leading to larger energy barrier in the low temperature range.

(4) The ions inside the 2D layers are still hydrated, or at least partially. Can the authors explain the mechanism of high conductivity of the ionic conductor under a temperature of -30 degree C?

Reply: We thank the reviewer very much for the insightful comments.

The Raman spectra of liquid water and ice are different due to the different water molecular arrangements. Typically, the O-H stretching vibration of water molecules shows a broad peak at 3000 – 3700 cm^{-1} , composed of various components related to different intermolecular bonding degrees of water [*Nat. Commun.* 11, 4463, 2020; *J. Phys. Chem. A* 118, 2922-2930, 2014]. Generally, the peak position decreases as the H bonds become stronger. The Raman spectrum of liquid water is dominated with weak H-bond peak. In contrast, ice shows a sharp Raman peak corresponding to the stretching vibration of strong H bonds. Analyzing the Raman peak position and proportion can provide insights into the arrangement state of water molecules in the material.

In our experiment, we studied the Raman spectra of CdPS₃-Li membranes at different temperatures. The proportion of strongly H-bonded water (at ~3230 cm^{-1}) greatly increases while non-H-bonded water (at ~3620 cm^{-1}) disappears as the temperature decreases below 0 °C for pure water (Supplementary Fig. 28, 29a, b), suggesting the formation of ordered ice. Surprisingly, the proportion of strongly H-bonded and non-H-bonded water molecules in CdPS₃-Li membranes only slightly increases and decreases, respectively, even down to -40 °C (Fig. 4e, f, Supplementary Fig. 29c, d). The H-bond is formed between the partially positively charged H atoms and partially negatively charged O atoms of neighboring H₂O mainly by electrostatic interaction [*Nat. Commun.* 11, 4463, 2020]. Taking into account the high-density cations confined within the 2D nanochannels, which contain at most bilayer water, we expect that the coordination of water molecules with cations would destroy the H-bond interactions between water molecules. The destruction of H-bonds enables the survival of non-H-bonded water molecules, which ensures high conductivity even down to -30 °C.

(5) The authors use concerted ion movement to explain the high mobility of ions inside the 2d channels. Can the enhancement of 1.6 times higher mobility of K⁺ be explained by the mean field theory with considering the Coulomb interaction of positive ions? In ref. 16, the mean field theoretical model predicts an enhancement around 40 times in graphene channels.

Reply: We thank the reviewer very much for the insightful comments.

In ref. 16, the authors used a 0.38 nm nanopore with a surface charge density $n = 1.8 \times 10^{14} \text{ cm}^{-2}$ to achieve an enhancement of ion mobility around 40 times in graphene channels. We obtained a ~15-fold enhancement for ion mobility with the interlayer spacing (0.65 nm) and surface charge density ($2 e/\text{nm}^2$) of our membranes by using the mean field theoretical model. The mean field theory provides an idealized prediction of ion mobility in a charged nanochannel under simplified conditions (free of defects and charge traps) [*Nat. Mater.* 13, 387-393, 2014]. However, the actual changes in the nanochannel composition, geometry, and structure can affect ion transport in real experiments, leading to discrepancies between the predicted and measured results. For instance, recent studies have shown low friction between water molecules/ions and the channel wall in carbon nanotubes, whereas in boron nitride nanotubes, which are crystallographically similar to carbon nanotubes, high friction is observed, indicating the crucial role of nanotube walls in particle transport [*Nature*, 567, 87-90, 2019; *Nature*,

537, 210-213, 2016; *Nat. Mater.* 19, 1057-1061, 2020]. Ref. 16 used graphene nanopores with a smooth surface, which are closely aligned with the mean-field model, resulting in good agreement between the predicted and measured results. Compared with carbon nanotubes or graphene channels, the nanochannels of CdPS₃ have much higher roughness and abundant charge traps (vacancies), which potentially reduces the actual ion mobility. In this scenario, the measured enhancement of 1.6 times higher mobility of K⁺ is reasonably consistent with the predictions by the mean field theory.

We have added the relevant discussion in the revised manuscript.

(6) In the XRD measurement of Al³⁺ intercalated samples, the peak is very weak. Does it mean the Al³⁺ is hardly intercalated inside the 2D channels? Do the lithium ions completely replaced by other ions during the immersion? How to prove that?

Reply: We thank the reviewer very much for the insightful comments.

Inductively coupled plasma-atomic emission spectroscopy (ICP-AES) is a sensitive analytical technique that can detect and quantify trace elements in a material. In our study, we used ICP-AES to identify the composition of the membranes. Table R1 below presents the test results of CdPS₃-Al membrane, which indicate that there are only Cd, P, S and Al in the membrane. Notably, the content of Li is below the detection limit of the ICP-AES, suggesting that the original Li⁺ ions present in the interlayer have been entirely exchanged by Al³⁺. The survey spectra of XPS with depth analysis also indicates that no lithium was detected (Fig. R10q).

Table. R1 Elemental composition of the CdPS₃-Al membrane determined by ICP-AES.

Element	Unit	Mean	Standard Deviation	%RSD
Cd	mg L ⁻¹	80.28	0.38	0.4695
P	mg L ⁻¹	26.2	0.08	0.2894
S	mg L ⁻¹	86.79	0.24	0.2819
Al	mg L ⁻¹	2.728	0.024	0.8887
Li	mg L ⁻¹	0.0119	0.0100	84.02

In Fig. S16, the CdPS₃-Al membrane shows a weak XRD peak at 7.9°, which corresponds to an interlayer distance of 1.12 nm. However, we should point out that this XRD pattern was measured under ambient conditions (~30-50% RH%). As shown in Fig. 1d, the CdPS₃-Al membrane shows similar strong XRD peak at 6.7° at 98% RH. Moreover, a similar exchange time is required to obtain different CdPS₃-Z membranes. Combining with the ICP-AES and XPS results shown above, we suggest that Al³⁺ ions are easily intercalated inside the 2D channels similar to other cations.

REVIEWERS' COMMENTS

Reviewer #1 (Remarks to the Author):

The authors have addressed all of my comments, and I recommend publishing this manuscript in Nature Communications.

Reviewer #2 (Remarks to the Author):

The authors have adequately addressed all the comments, and the revised version is acceptable.

Reviewer #3 (Remarks to the Author):

The responses are satisfying, and I have no further comments.

Response to reviewers' comments

Reviewer #1 (Remarks to the Author):

The authors have addressed all of my comments, and I recommend publishing this manuscript in Nature Communications.

Response: We thank the reviewer very much for the positive comments.

Reviewer #2 (Remarks to the Author):

The authors have adequately addressed all the comments, and the revised version is acceptable.

Response: We thank the reviewer very much for the positive comments.

Reviewer #3 (Remarks to the Author):

The responses are satisfying, and I have no further comments.

Response: We thank the reviewer very much for the positive comments.